# LaGrACE: estimating gene program dysregulation with latent regulatory network

Minxue Jia [1,2], Haiyi Mao[1,2], Mengli Zhou[1,3], Yu-Chih Chen[1,2,3] & Panayiotis V Benos [1,2,4✉]

## Abstract

**Gene expression programs that establish and maintain specific cellular states are orchestrated through a regulatory network composed of transcription factors, cofactors, and chromatin regulators. Dysregulation of this network can lead to a broad range of diseases by altering gene programs. This article presents LaGrACE, a novel method designed to estimate dysregulation of gene programs combining omics data with clinical information. This approach facilitates the grouping of samples exhibiting similar patterns of gene program dysregulation, thereby enhancing the discovery of underlying molecular mechanisms in disease subpopulations. We rigorously evaluated LaGrACE's performance using synthetic data, bulk RNA-seq clinical datasets (breast cancer, chronic obstructive pulmonary disease (COPD)), and single-cell RNA-seq drug perturbation datasets. Our findings demonstrate that LaGrACE is exceptionally robust in identifying biologically meaningful and prognostic molecular subtypes. In addition, it effectively discerns drug response signals at a single-cell resolution. Moreover, the COPD analysis uncovered a new role of LEF1 regulator in COPD molecular mechanisms associated with mortality. Collectively, these results underscore the utility of LaGrACE as a valuable tool for elucidating the underlying mechanisms of diseases.**

**Keywords** Causal Graphs; Gene Programs; COPD
**Subject Categories** Chromatin, Transcription & Genomics; Computational Biology

## Introduction

Gene programs are sets of genes that facilitate transcriptional regulation mechanisms and collectively contribute to a specific biological function under particular conditions or phenotypes (Harris et al, 2021). Gene programs can interact and work cooperatively through complex mechanisms. There are a number of factors that affect the gene programs and the complex mechanisms they participate, like transcription factor (TF) activities, protein signaling, chromatin accessibility or microRNA expression (Lambert et al, 2018). In most typical datasets, multiple of these modalities are not observed. Furthermore, the control that these factors exert on the gene programs and their interactions creates collinearity in the expression of the member genes. Patient phenotypes may be the result of differences in these gene programs and their connections, reflecting their roles in clinical outcomes, responses to treatments, or molecular disease subtypes. Although these factors are not directly observable, they provide a more comprehensive picture of the genetic mechanism of disease and facilitate the development of effective therapeutic strategies.

To harness this potential, significant advances have been made in methods that quantify gene program activity on a single patient sample or individual cell through unsupervised factorization approaches, such as non-negative matrix factorization (NMF). But most methods are not generally identifiable (Pan and Doshi-Velez, 2016; Kotliar et al, 2019; Brunet et al, 2004), which hampers the interpretability of the identified factors. Moreover, some orthogonal factor models, like PCA, DIALOGUE, and scITD (Jerby-Arnon and Regev, 2022; Mitchel et al, 2024), overlook the fact that gene programs are organized in networks that collectively control the observed expression of their member genes. Conversely, considerable progress has been made in methods that quantify gene program activity by leveraging existing information from curated databases. Examples of such methods include Spectra (Kunes et al, 2023), expiMap (Lotfollahi et al, 2023), Pathifier (Drier et al, 2013), and ssGSEA (Barbie et al, 2009). Nevertheless, these approaches strongly depend on the curated gene programs, which are derived from multiple biological contexts, and may not correspond to the specific context under study.

Those limitations become particularly evident when considering the dynamic nature of gene regulation in response to treatment or disease (Lee and Young, 2013). Such alterations can lead to divergent changes in functionally relevant gene programs if shared regulatory elements (e.g., bound TFs and mediators) are modified across conditions (Fig. 1A). Measuring alterations in the gene programs regulatory network is challenging. A primary challenge in biomedical research is the inherent inability to observe identical samples simultaneously in both treatment and control states. Motivated by our understanding of gene regulatory networks, we introduce Latent Graph-based individuAl Causal Effect Estimation (LaGrACE). LaGrACE is a novel approach designed to estimate regulatory network-based pseudo-control outcome to characterize gene program dysregulation for samples within treatment (or disease) group. This method enables grouping of samples with

[1]Department of Computational and Systems Biology, University of Pittsburgh School of Medicine, Pittsburgh, PA, USA. [2]Joint CMU-Pitt PhD Program in Computational Biology, Pittsburgh, PA, USA. [3]UPMC Hillman Cancer Center, Pittsburgh, PA, USA. [4] Department of Epidemiology, University of Florida, Gainesville, FL, USA. ✉E-mail: pbenos@ufl.edu

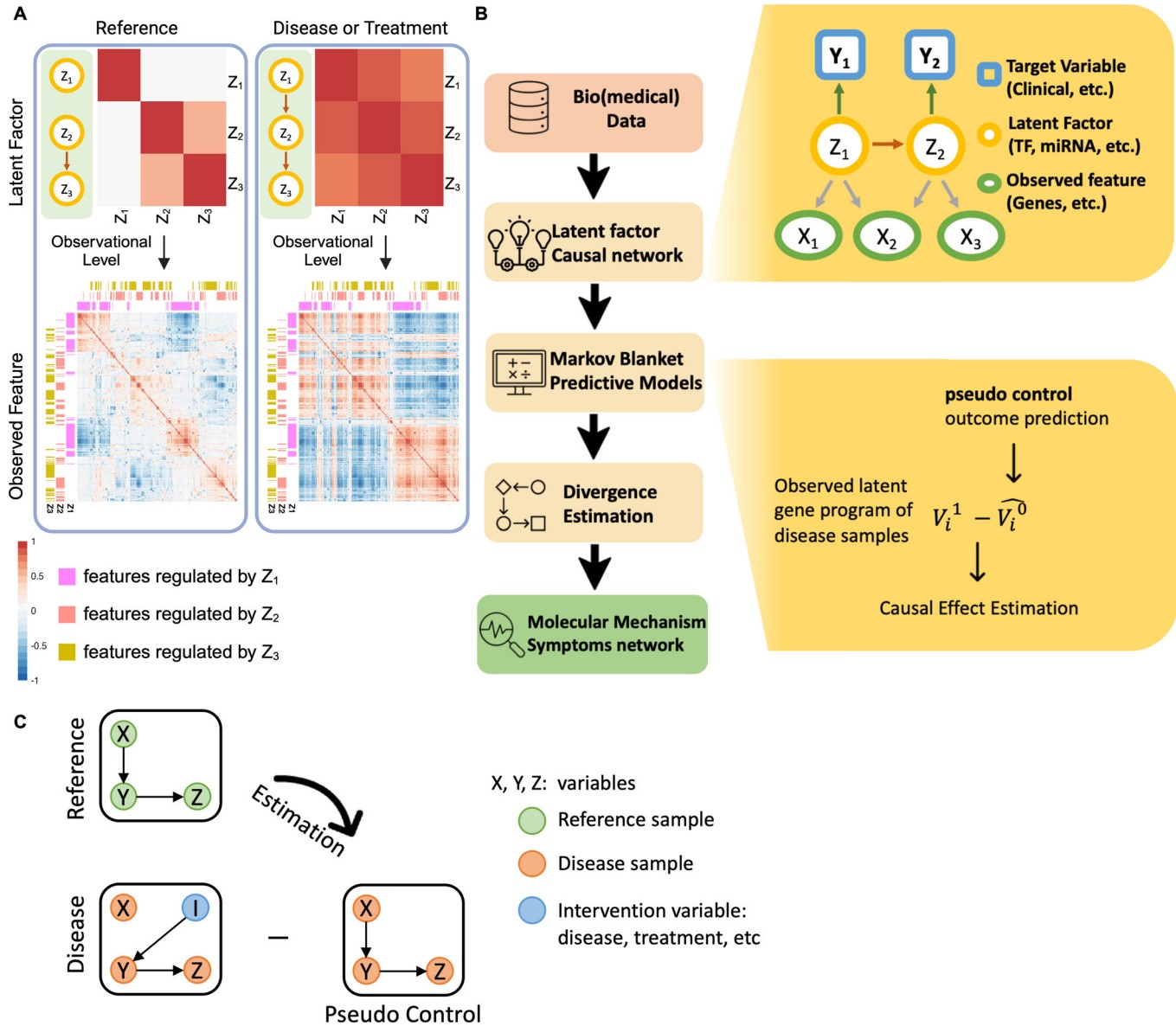

**Figure 1. Change in gene program interactions and algorithm overview of LaGrACE.**

(A) Three latent factors (LFs) regulate a set of observed features separately in the reference condition (left). If interactions among three LFs were changed in disease (or treatment) condition (right), the correlation of regulated features would be altered consequently. (B) LaGrACE is a framework to construct pseudo-control leveraging regulatory network of reference sample and quantify heterogeneity of disease group through divergence estimation. (C) Illustration of leveraging pseudo-control outcome to calculate divergence score.

similar patterns of gene program dysregulation, thereby facilitating the discovery of underlying molecular mechanisms induced by treatment or disease.

# Results

## LaGrACE infers divergence score for gene program dysregulation

The gene programs, that collectively engage in the same biological function, often interact with each other (Kunes et al, 2023).

Perturbation in those gene programs and their interactions can lead to a cascade of dysregulation, affecting gene expression, pathways, and other cellular processes. A significant challenge arises in modeling alterations in gene programs within a single sample, especially when a biomedical sample can typically only be observed in one condition at a time, either the control or the noncontrol (treatment) state. This is a common issue in biomedical studies.

We propose a novel approach to estimate the single-sample changes in gene program interactions compared to a reference gene program network. First, we start by assuming that there is a Bayesian network, delineating gene program interactions present in all control samples ("reference network") (Segal et al, 2003). In any

given disease (or treatment) sample, we expect that some of the gene programs and their interactions will change. To assess these changes in this sample, we do the following. In the "reference network", we build a predictor of a gene program activity by using the variables in its Markov blanket (i.e., the parents, children, and spouses; see "Methods"). Second, we measure the difference between the expected value of each gene program based on the "reference network" and the observed values in the (treatment or disease) sample. Specifically, we assume that the general structure of this "reference network" for the most part is maintained in samples from the disease (or treatment) group, although certain part of this network may be altered in the disease (or treatment) samples. The altered parts will depend on the particular characteristics of the patient. This approach allows for the assessment of sample-level dysregulation of the (unobserved) gene programs in disease (or treatment) samples, based on omics profiling and clinical features (see Fig. 1B). But this raises a question: if a given disease sample could also be observed in a control state, will gene programs with analogous biological functions exhibit comparable behavior?

In a clinical study, we can only measure transcriptomic profiles in a sample once: either in the control or in the disease (or treatment) state, leaving the other side unobserved. Let W (1: treated; 0: untreated) represent the treatment label assignment variable for an individual $j$, where only a portion of the variables can be directly observed. Let $V_{ij}$ denote the observed feature $i$ (gene program or clinical variable) from individual $j$ and '?' denotes an unobserved state:

$$V_{ij}^0 = \begin{cases} V_{ij}, W_j = 0 \\ ?, W_j = 1 \end{cases}, V_{ij}^1 = \begin{cases} ?, W_j = 0 \\ V_{ij}, W_j = 1 \end{cases}$$

To estimate control state ($W_j = 0$) of a disease (or treated) sample $V_{ij}^1$, we calculate the pseudo-control outcome $\hat{V}_{ij}^0$, leveraging Markov blanket of $V_i$ derived from the "reference network" (Fig. 1C). This approach assumes that gene programs with analogous biological functions in control samples maintain similar behavior in disease samples.

Before estimating the pseudo-control outcome, we first discuss how to estimate gene programs. We infer gene programs from high-dimensional gene expression data. Gene programs are typically estimated using factor analysis or NMF methods. However, these methods often face issues with identifiability and impose orthogonality constraints. Here, we utilize the LOVE algorithm (Bing et al, 2019), which addresses these limitations by providing (1) theoretical guarantees regarding the unique identifiability of latent factors without relying on assumptions about the data-generating mechanisms, and (2) the ability to identify non-orthogonal latent factors without restrictive assumptions regarding orthogonality.

Then, we infer a reference network, represented as a Bayesian network, to delineate interactions among gene programs (Z) and between gene programs and clinical variables (U). With such a reference network inferred from reference (or control) samples, the value of a gene program $Z_i$ can be approximated based on its Markov Blanket $MB(Z_i)$ using a regression model. A set of Markov Blanket regression models, trained on reference samples, represents reference gene program interactions. The pseudo-control outcome of treatment samples, $\hat{Z}_i^0$, can then be estimated using those trained

regression models based on reference samples. If the Markov Blanket of a gene program is altered under treatment condition, this alteration can be quantified by calculating expectation divergence, which is the difference between observed value and the pseudo-control outcome in treatment samples. For variable $V_i$ (representing gene program $V_i$ or clinical variable $U_i$) in sample $j$,

$$\text{Divergence Score}: V_{ij}^1 - \hat{V}_{ij}^0$$

These divergence scores can then be used to group samples based on patterns of dysregulation by evaluating alterations within the reference network and their magnitudes.

Changes in the gene program regulation network, induced by disease or treatment, result in phenotypic changes, including mortality and morphological transformations. Notably, identical phenotype or outcomes associated with a disease or treatment can be observed as a result of changes in different parts of this network. For instance, patients with Alzheimer's disease or cancer can be stratified into molecular subtypes characterized by distinct dysregulated pathways (Neff et al, 2021; Rosario et al, 2018). In this work, we evaluate LaGrACE based on the ability to identify such subtypes.

## LaGrACE effectively recovers network subtypes in low- and high-dimensional synthetic data

Before applying LaGrACE to high-dimensional, collinear datasets for subtype identification, we conducted initial evaluations on low-dimensional synthetic datasets. Each synthetic dataset consisted of a reference network alongside three perturbed networks, each subjected to random edge alterations (both deletions and additions). These modifications simulate variations corresponding to disease subtypes, treatment categories, or clinical interventions (see "Methods"). All simulated networks consisted of 50 continuous and 25 categorical variables that resemble the variable types observed in biomedical datasets. Each network was used to generate 1000 samples. LaGrACE learned the reference network from the reference network samples. Subsequently, we calculated LaGrACE divergence scores for each "disease" sample, generated by one of the perturbed networks. We clustered the samples based on the divergence scores (see "Methods"). The resulting clusters demonstrated improved separation and better Adjusted Rand Index (ARI) and Adjusted Mutual Information (AMI) scores compared to observed continuous variables from synthetic data (Fig. 2A,B). These results suggest LaGrACE's superior performance in subtype delineation. This experiment was repeated with smaller reference sample sizes, 100 and 500 samples randomly drawn from 1000 original samples. Even with fewer reference samples, LaGrACE's divergence estimation method continued to accurately identify distinct clusters (Fig. 2C).

Next, we evaluated LaGrACE on high-dimensional synthetic datasets, comprising 25 discrete and 2500 continuous variables generated from simulated latent factors (see "Methods"), which represent clinical features (discrete) and gene expression (continuous). The perturbed groups delineated by LaGrACE divergence score demonstrated better separation compared to observed continuous variables' (Fig. 2D). We clustered the samples using (1) observed continuous variables, (2) LaGrACE divergence with ground truth latent factors, (3) LaGrACE divergence with LOVE-recovered latent factors (LOVE LF) and discrete variables, (4)

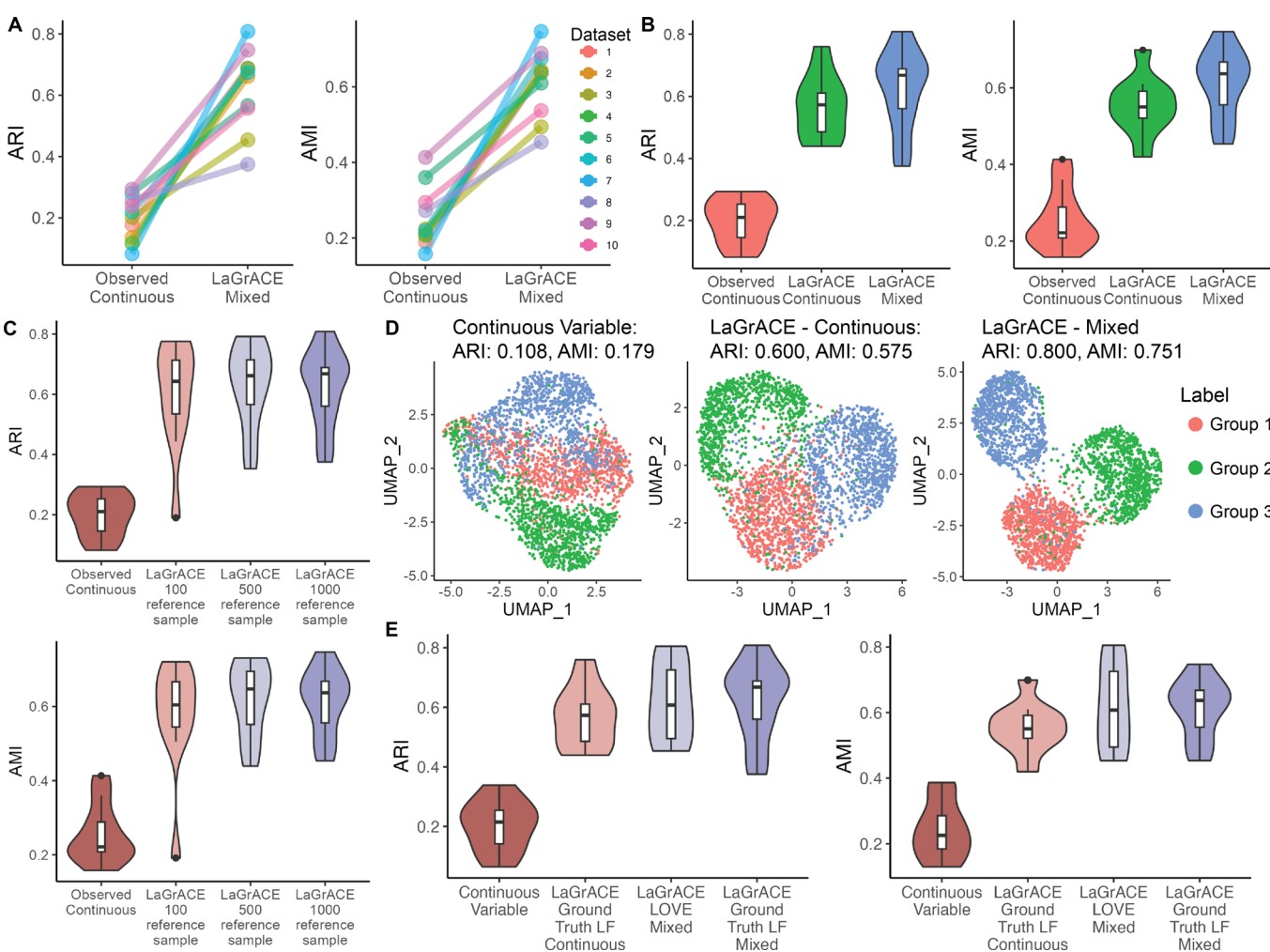

**Figure 2. Evaluation of LaGrACE's clustering performance on synthetic dataset with 15% edge change.**

(A) Slope chart of ARI and AMI of clustering based on 50 continuous variables (1st point) and LaGrACE feature inferred from both 50 continuous variables and 25 discrete variables (2nd point). (B) Boxplot of ARI and AMI of clustering based on simulated 50 continuous variables (1st point, $n = 10$), LaGrACE feature inferred from continuous variables only (2nd point, $n = 10$) and LaGrACE feature inferred from both continuous variables and discrete variables (3rd point, $n = 10$), where n represents the number of synthetic datasets. (C) Violin plot of ARI and AMI of clustering based on 50 continuous variables and LaGrACE features with 100, 500 and 1000 reference samples. For each group, $n = 10$. (D) UMAP plot of simulated data colored by clusters based on high-dimensional (2500) continuous variables (ARI: 0.108; AMI:0.179), LaGrACE features inferred from continuous variables only (ARI: 0.600; AMI:0.575) and LaGrACE feature inferred from both continuous variables and discrete variables (ARI: 0.800, AMI: 0.751). (E) Violin plot of ARI and AMI of clustering based on high-dimensional continuous observed variables, LaGrACE feature inferred from ground truth LF, LaGrACE features inferred from LOVE LF and discrete variables, and LaGrACE feature inferred from ground truth LF and discrete variables. For each group, $n = 10$. Box plots indicate the median (50th percentile), the lower (25th percentile) and upper (75th percentile) quartiles. The whiskers extend to the minimum and maximum values within 1.5 times the interquartile range (IQR) below the 25th percentile and above the 75th percentile, respectively. Data points beyond 1.5×IQR and less than 3×IQR are shown as whiskers, while those beyond 3×IQR are considered extreme outliers.

LaGrACE divergence with ground truth latent factors and discrete variables (Fig. 2E). Notably, LaGrACE divergence score achieved best performance. Also, LaGrACE based on LOVE LF exhibited performance comparable to LaGrACE with ground truth latent factors. This indicates that LaGrACE is robust for subtyping tasks in high-dimensional and collinear datasets.

## LaGrACE demonstrates precise drug response detection at the single-cell level

We also tested LaGrACE's performance in a number of real-life datasets. scRNA data were collected from A549 lung adenocarcinoma cells treated with dexamethasone over intervals of 0, 1, and 3 h (Cao et al, 2018). Addressing scRNA-seq data sparsity, we utilized imputed RNA features estimated by scVI (Lopez et al, 2018) as input for LaGrACE. The vehicle group served as a reference for network inference, Markov blanket estimators, and divergence score computation. LaGrACE divergence score facilitated clear stratification of cells by treatment duration in UMAP embedding, surpassing the classification achieved using (unimputed and imputed) RNA features and gene set scores inferred using Spectra and expiMap (Fig. 3A,B). Higher ARI, AMI, and treatment time purity scores in LaGrACE clusters underscored its capacity at processing single-cell level data (Fig. 3C). Association

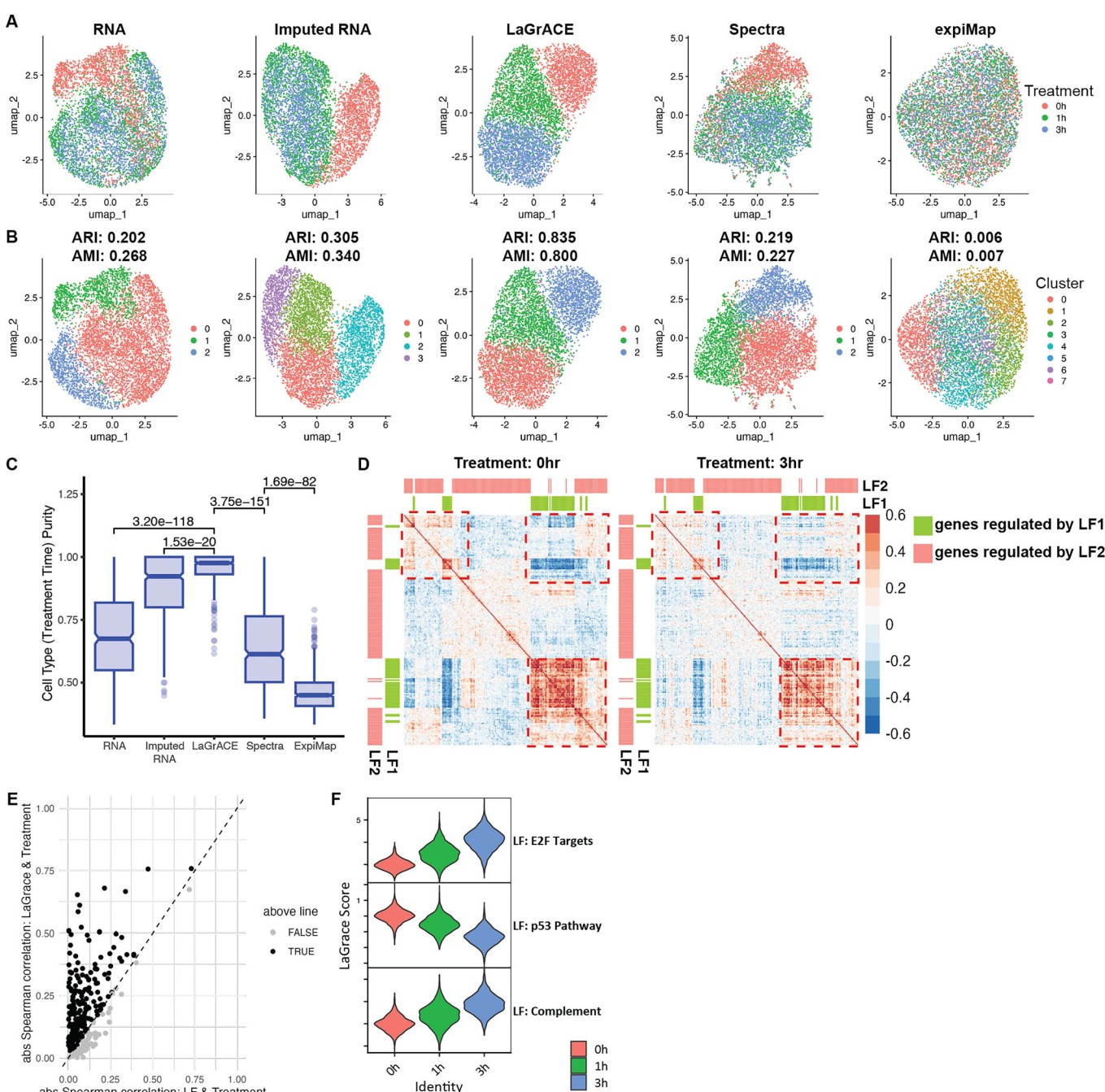

**Figure 3. LaGrACE captures treatment effect at single-cell resolution.**

A549 lung adenocarcinoma cells were treated with 100 nM dexamethasone for 1 h or 3 h or ethanol (control group). (A) A549 cells visualized and colored by treatment time on UMAP embeddings computed based on unimputed RNA profiles, RNA profiles imputed by SCVI, LaGrACE features and gene set scores inferred by Spectra and expiMap. (B) UMAP plot of A549 cells colored by clusters based on unimputed RNA (ARI: 0.202, AMI: 0.268), imputed RNA (ARI: 0.305, AMI: 0.340), LaGrACE features (ARI: 0.835, AMI: 0.800), Spectra gene set scores (ARI: 0.219, AMI: 0.227), and expiMap gene set scores (ARI: 0.006, AMI: 0.007). (C) Boxplot visualized purity score of cell type (treatment time) for single-cell Neighborhoods constructed using Milo, based on RNA ($n = 383$), imputed RNA ($n = 506$), LaGrACE ($n = 387$), Spectra ($n = 465$), and expiMap ($n = 397$), where n represents the number of single-cell neighborhoods (P values: Student's t test). Box plots indicate the median (50th percentile), the lower (25th percentile) and upper (75th percentile) quartiles. The whiskers extend to the minimum and maximum values within 1.5 times the interquartile range (IQR) below the 25th percentile and above the 75th percentile, respectively. Data points beyond 1.5×IQR and less than 3×IQR are shown as whiskers, while those beyond 3×IQR are considered extreme outliers. (D) Correlation heatmaps of genes regulated by LF1 and its Markov blanket (LF2) at different treatment time points. The region highlighted by the dashed line box exhibits change in correlation of gene expression regulated by LF1 and its Markov Blanket (LF2) across conditions. (E) Scatter plot of absolute Spearman rank correlation coefficient between treatment time and latent factors (X axis)/LaGrACE scores of latent factors (Y axis). (F) Violin plot of three LaGrACE features significantly correlated with treatment time (spearman correlation coefficient of LF: E2F targets, LF: p53 Pathway and LF: Complement: 0.758, −0.756 and 0.666), based on a cell population treated with dexamethasone for 1 h ($n = 2037$), 3 h ($n = 2269$), or control ($n = 1646$). Source data are available online for this figure.

among gene expression features, regulated by LF1 and its Markov Blanket LF2, varied at different treatment conditions (Fig. 3D), leading to changes in gene programs. Strikingly, divergence scores for LOVE latent factors exhibited higher Spearman correlation with treatment duration compared to the latent factors themselves (Fig. 3E). This indicates that LaGrACE effectively discerned gene program dysregulation.

Three latent factors with divergence scores exhibiting significant correlations with treatment duration were identified. These factors were significantly enriched in pathways related to E2F targets, the p53 pathway, and the complement system (Fig. 3F). Such findings align with known dexamethasone effects on p53 phosphorylation, E2F transcriptional activity and synthesis of complement proteins in A549 cells (Greenberg et al, 2002; Urban et al, 2003; Hill et al, 1993).

Further application of LaGrACE to an additional scRNA-seq dataset, A549 cells treated with small molecules (BMS-345541, nutlin-3A, vorinostat) over a 24-h period and across various doses (Srivatsan et al, 2020), showed that our approach adeptly captured dose-responsive signals. This was evident from UMAP embeddings, clustering accuracy (ARI and AMI values), and cell-type purity scores inferred from drug dose information, surpassing the performance of gene expression features and prior knowledge-based gene set scores (Appendix Figs. S1–S3).

## New breast cancer molecular subtypes identified and validated by LaGrACE

Application of LaGrACE to the METABRIC breast cancer dataset uncovered potential new molecular subtypes. In the absence of normal tissue samples in METABRIC, basal subtype samples, which is significantly different from all other molecular phenotypes, was used as the reference group. Figure 4A–D shows the clustering separation using the standard molecular subtypes, and the disease-specific survival, time to distant and loco-regional relapse for these subtypes. Utilizing 1368 latent factors from RNA-seq data, along with clinical features (i.e., ER/HER2/PR status, positive lymph nodes, age at diagnosis, and tumor size), LaGrACE identified six clusters (Fig. 4E–H; Table 1) significantly associated with molecular subtypes (P value < 2.2e-16, Chi-square test), disease-specific survival (P value = 1.31e-22), and distant (P value = 1.91e-21) and loco-regional relapse (P value = 1.45e-5), and clinical features (Appendix Table S1). Cluster 0 notably grouped 82% of HER2+ and 29% of luminal B samples (Table 1). Clusters 1, 2, and 3 were mainly luminal samples, with higher proportions of luminal A in cluster 1 and luminal B in cluster 3. Claudin-low samples were primarily found in Clusters 4 and 5. Clusters 1, 2, and 3, all luminal, showed varying survival rates, metastasis rates, and other clinical features (Appendix Table S1 and Appendix Fig. S4). with Cluster 1 showing better survival and lower metastasis rates compared to Clusters 2 and 3. Cluster 2 was differentiated from Cluster 3 by menopausal states, tumor cellularity, and the Nottingham Prognostic Index. The Claudin-low Clusters 4 and 5 exhibited variations in cancer histology and menopausal states, highlighting our method's ability to distinguish different clinical phenotypes within similar molecular subtypes (Appendix Fig. S5A–C). Further analysis of the clinical features associated with luminal B and HER2+ samples across LaGrACE clusters provided additional insights into their separation into multiple clusters (Appendix Fig. S5D–G).

For validation, cluster memberships were assigned to new samples from an external breast cancer dataset (SCAN-B) based on their divergence scores. The clusters assigned to SCAN-B samples were significantly associated with molecular subtypes (P value < 2.2e-16), survival (P value = 1.61e-07) and clinical features (Fig. 4I,J; Appendix Table S1). Following the same trend as in METABRIC, most HER+ samples were in cluster 0 (Table 1). Cluster 1 had the highest survival rates, and cluster 2 exhibited better histological scores than cluster 3 (Appendix Figs. S4 and S5 and Appendix Table S2). No samples were assigned to cluster 5. Notably, the METABRIC patients were enrolled before trastuzumab availability (anti-HER2 treatment), leading to a higher hazard ratio in Cluster 0 compared to that observed in the SCAN-B dataset.

Comparison with prior knowledge-based methods (pathway information) yielded similar results for molecular subtypes within the METABRIC dataset (Appendix Fig. S6). Single-sample Gene Set Enrichment Analysis (ssGSEA) and Pathifier identified 7 and 9 clusters, respectively, which were significantly associated with molecular subtype, survival, and metastasis (P values: <2.2e-16, 4.22e-16, 6.33e-14 for ssGSEA; <2.2e-16, 7.06e-9, 1.02e-7 for Pathifier). Employing the linear predictor derived from the Cox proportional hazard model, we evaluated the discriminatory power of identified clusters using Harrell's C-index and the concordance probability estimate (CPE). LaGrACE outperformed both ssGSEA and Pathifier in prognostic value for survival and recurrence (Appendix Table S3). Time-dependent AUC analysis also showed LaGrACE's superior discrimination ability in metastasis, survival, and local relapse in both METABRIC and SCAN-B datasets (Fig. 4K). Overall, LaGrACE achieved great performance in classifying tumor samples by molecular subtype, survival, and disease progression.

## Identification of biological networks driving breast cancer subtypes networks

LaGrACE clusters were instrumental in analyzing molecular alterations impacting clinical phenotype and progression of breast cancer. Epithelial cell differentiation score(Prat et al, 2010), proliferation index (Nielsen et al, 2010), and immune cytolytic activity(Rooney et al, 2015) were independently associated with LaGrACE subtypes (Appendix Fig. S7A–C). The luminal clusters 1, 2, 3 demonstrated varying scores: Cluster 1 exhibited lower differentiation and proliferation rates but higher cytolytic activity. Conversely, clusters 2 and 3 presented higher differentiation and lower cytolytic activity, with cluster 3 exhibiting increased proliferation rates. Claudin-low clusters 4 and 5 were characterized by a high inferred immune cellular fraction (Appendix Fig. S8), with Cluster 4 expressing high levels of immune exhaustion markers PD-1 and LAG3 (Appendix Fig. S7F). This indicates an immunosuppressive environment and early tumor progression (Rye et al, 2022).

LaGrACE subtypes allow us to develop subtype-specific networks, integrating clinical variables and latent factors to elucidate disease-associated molecular mechanisms. We used the FCI-Stable algorithm to infer networks for each luminal cluster (Appendix Fig. S9). Focusing on tumor size-related variables, all luminal cluster networks found lymph node number to be directly linked to tumor size, but different latent factors were associated with tumor

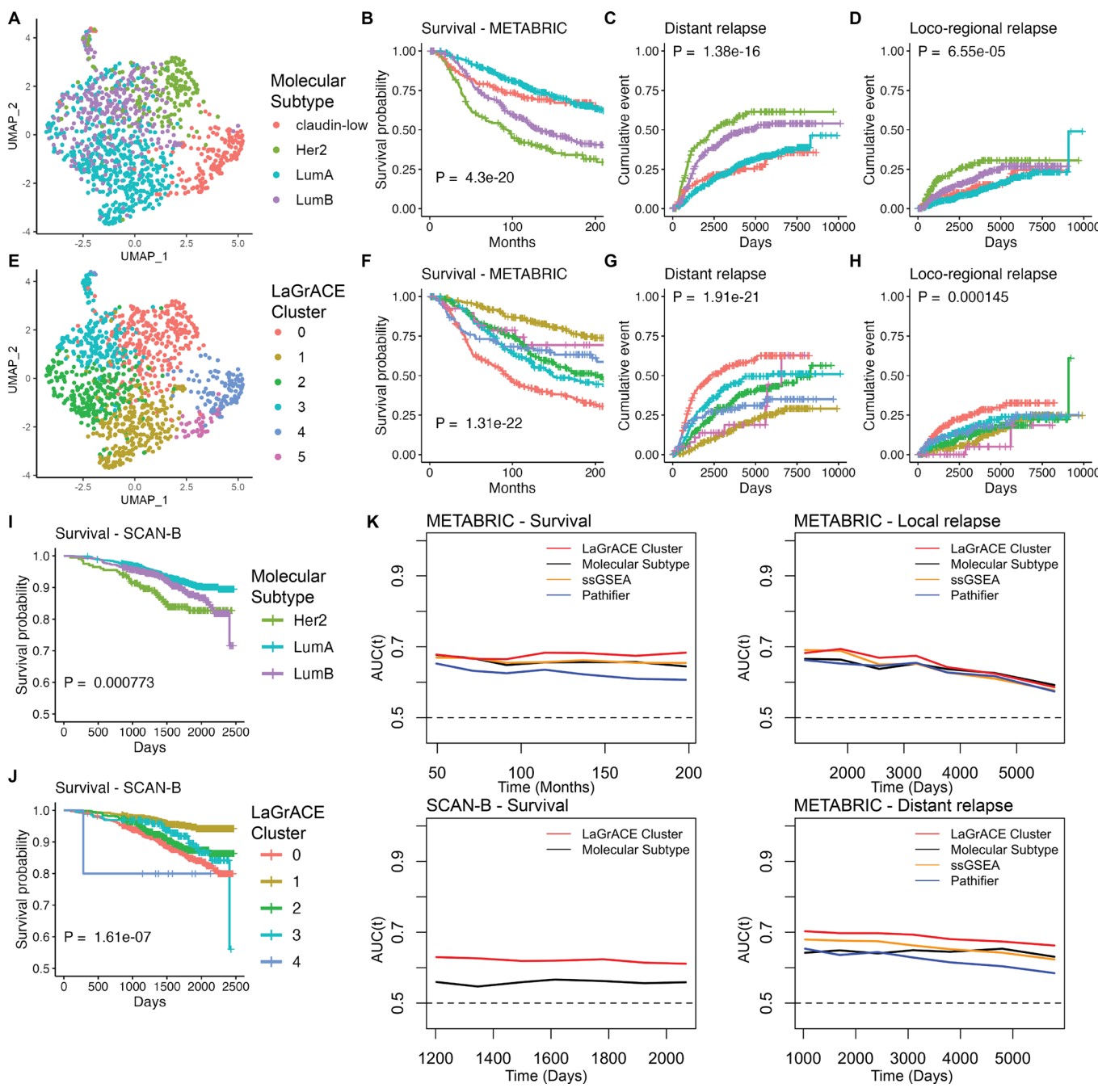

**Figure 4.   LaGrACE identified six clusters on the breast cancer dataset.**

(**A**) UMAP visualization of METABRIC breast cancer samples colored by tumor molecular subtype. (**B**) Breast cancer survival curve for molecular subtypes on METABRIC dataset. (**C**) Kaplan–Meier estimate of distant relapse on METABRIC dataset by molecular subtype. (**D**) Kaplan–Meier estimate of local-regional relapse on METABRIC dataset by molecular subtype. (**E**) UMAP visualization of METABRIC breast cancer samples colored by LaGrACE clusters. (**F**) Breast cancer survival curve for LaGrACE clusters on METABRIC dataset. (**G**) Kaplan–Meier estimate of distant relapse on METABRIC dataset by LaGrACE clusters. (**H**) Kaplan–Meier estimate of local-regional on METABRIC dataset by LaGrACE clusters. (**I**) Breast cancer survival curve for molecular subtypes on SCAN-B dataset. (**J**) Breast cancer survival curve for LaGrACE clusters on SCAN-B dataset. (**K**) Time-dependent AUC of survival and recurrent event on samples from METABRIC and SCAN-B dataset. The *P* values in the Kaplan–Meier survival (or relapse) analysis were calculated using the log-rank test. Event definitions. Survival event (1 = YES, 0 = NO); distant relapse (1 = YES, 0 = NO); loco-regional relapse (1 = YES, 0 = NO); time until last follow-up or death (in months); time until last follow-up or distant relapse (in days); Time until last follow-up or loco-regional relapse (in days).

**Table 1. Molecular subtype composition of LaGrACE breast cancer identified clusters.**

| | METABRIC | | | | SCAN-B | | |
|---|---|---|---|---|---|---|---|
| | Claudin-low | Her2 | LumA | LumB | Her2 | LumA | LumB |
| 0 | 4 | 140 | 36 | 97 | 193 | 228 | 284 |
| 1 | 15 | 3 | 216 | 28 | 2 | 741 | 27 |
| 2 | 0 | 8 | 155 | 79 | 0 | 354 | 153 |
| 3 | 2 | 10 | 53 | 129 | 2 | 47 | 148 |
| 4 | 102 | 9 | 0 | 4 | 4 | 2 | 4 |
| 5 | 41 | 0 | 0 | 0 | 0 | 0 | 0 |

size in each subtype. In cluster 1, a latent factor linked to tumor size is related to the apical junction, which stabilizes cell–cell adhesion and cell polarity. Metastasis is characterized by loss of cell–cell adhesion and cell polarity (González-Mariscal et al, 2020). This may explain the low metastasis rate in cluster 1 due to its significantly higher pathway enrichment score in junction-associated pathways, compared to clusters 2 and 3 (Appendix Fig. S7D).

For clusters 2 and 3, tumor size was linked to latent factors associated with by E2F transcriptional factors, which have been linked to the aggressiveness of breast cancer(Johnson et al, 2016). Cluster 1 exhibited downregulation in E2F target genes, including FOXM1, AURKB, and PLK1 (Appendix Fig. S7E). Those genes promote the WNT/β-catenin signaling pathway, crucial in breast cancer progression (Xu et al, 2020; Wang et al, 2020). We validated this finding by inhibiting FOXM1 in breast cancer cell lines with three different small-molecule FOXM1 inhibitors, and we observed reduced cell motility (Appendix Fig. S10). This illustrates that LaGrACE can be used to prioritize and highlight informative biomarkers and pathways within specific disease subtypes.

Further analysis of cluster-specific survival factors revealed distinct pathways associated with survival within each LaGrACE-defined cluster (Appendix Table S4). Galactose metabolism was significantly linked to survival in Cluster 1, aligning with its role in breast cancer survival and metastasis(Young et al, 2023; Han et al, 2023). Integrin-linked kinase (ILK) signaling was specifically associated with survival in Cluster 2, reflecting its relevance to metastatic progression and therapeutic vulnerabilities (McDonald and Dedhar 2022; Hinton, Avraham, and Avraham 2008; Beetham et al, 2022). For Cluster 3, E2F targets and the regulation of RhoA activity were strongly associated with survival, highlighting their contributions to cancer aggressiveness and metastasis(Mohammadalipour et al, 2022; Humphries, Wang, and Yang 2020). These findings emphasize the utility of LaGrACE in elucidating disease-associated molecular mechanisms tailored to each subgroup, providing valuable insights into breast cancer heterogeneity.

## Identification by LaGrACE of LEF1 and as an important COPD mortality factor via LaGrACE

We further applied LaGrACE to blood-derived RNA-sequencing gene expression measurements from 1750 subjects from COPD-Gene study, which is a longitudinal study that aims to discover the genetic basis of COPD disease mechanism. The reference group was meticulously selected based on following criteria: participants (a)

had clinical data from both phase 1 (baseline) and phase 2 (5-year follow-up) visits; (b) had no missing data for forced expiratory volume in 1 s (FEV1) and forced vital capacity (FVC); (c) had no spirometric abnormalities (i.e., GOLD 0) in phase 2 and phase 3 (10-yr follow-up) visits; (d) were not current smokers; (e) exhibited less than 5% percent emphysema in both phase 2 and phase 3 visits; (6) had less than 5% decrease in FEV1%predicted between phase 2 and phase 3 visits. This filtering procedure resulted in 139 reference and 1611 COPD subjects (former and current smokers).

Utilizing 602 latent factors from RNA-seq data and gender information, LaGrACE identified four clusters. Some clinical variables were significantly different between clusters, including spirometry, chest CT scan, symptom questionnaires, white blood cell differentials as well as smoking status (Fig. 5A,B; Table 2; Appendix Fig. S11). Cluster 0 represented the most impaired group among the three clusters that have predominantly non-Hispanic white (NHW) participants (Fig. 5B; Appendix Fig. S11B; Table 2), showing poorer spirometry results (FEV1/FVC ratio, FEV1% predicted), lower 6-min walking distance, generally higher symptom scores (measured by COPD Assessment Test and SGRQ St. George's Respiratory Questionnaire) and higher dyspnea severity (measured by Medical Research Council Dyspnea Scale). Cluster 3, which consists of predominantly African American (AA) participants, also has poorer FVC %predicted, lower 6-min walking distance, and higher SGRQ and dyspnea scores (Fig. 5B; Appendix Fig. S11B; Table 2). The neutrophil-to-lymphocyte ratio (NLR), a valuable predictor of acute exacerbations of COPD and mortality (Paliogiannis et al, 2018), displayed distinctive patterns in cluster 3 (Table 2; Appendix Fig. S12), which has more current smoking participants, higher lymphocyte and lower neutrophile level. This variation may be attributed to racial differences in leukocyte subpopulations and immune response (Nédélec et al, 2016). Cluster 1 and 2 showed comparable FEV1/FVC ratio and 6-min walking distance (Fig. 5B; Appendix Fig. S11B). Cluster 1 had lower neutrophil and CD4 naive T-cell percentage, but higher lymphocyte and CD4 resting memory T-cell percentage compared to cluster 2 (Appendix Fig. S12). In addition, the identified clusters were significantly associated with mortality (Fig. 5C), with cluster 1 maintaining the best survival probability and cluster 0 exhibiting the worst survival probability. CD4 naive and CD4 resting memory T cells, exhibiting significant differences across the four clusters, were prognostic for white patients but not for African American patients (Appendix Fig. S13A,B).

Two latent factors (Fig. 5D,E), enriched in lymphocyte percentage and CD4-specific genes, correlated significantly with lymphocyte parentage (measured from complete blood count test) and CD4 T cells (sum of CD4 naive T-cell and memory T-cellular fractions). Their LaGrACE divergence scores were significantly correlated with FEV1/FVC ratio and FEV1%predicted (Table 3). Unlike cluster 0 and 3, Cluster 1 showed no dysregulation on CD4 T and lymphocyte latent factors (Fig. 5F). Based on the inferred reference network, the two latent factors were linked to LEF1 motif (Fig. 5G). The transcriptional factor LEF1, which binds to β-catenin (CTNNB1) and regulates lymphopoiesis via canonical WNT signaling, also promotes commitment to the CD4 lineage through induction of Th-POK (van de Wetering et al, 2002; Steinke et al, 2014). Thus, we evaluated the association between LEF1 with COPD mortality. LEF1 transcriptome emerged as a risk factor for overall survival, as demonstrated through multivariate Cox

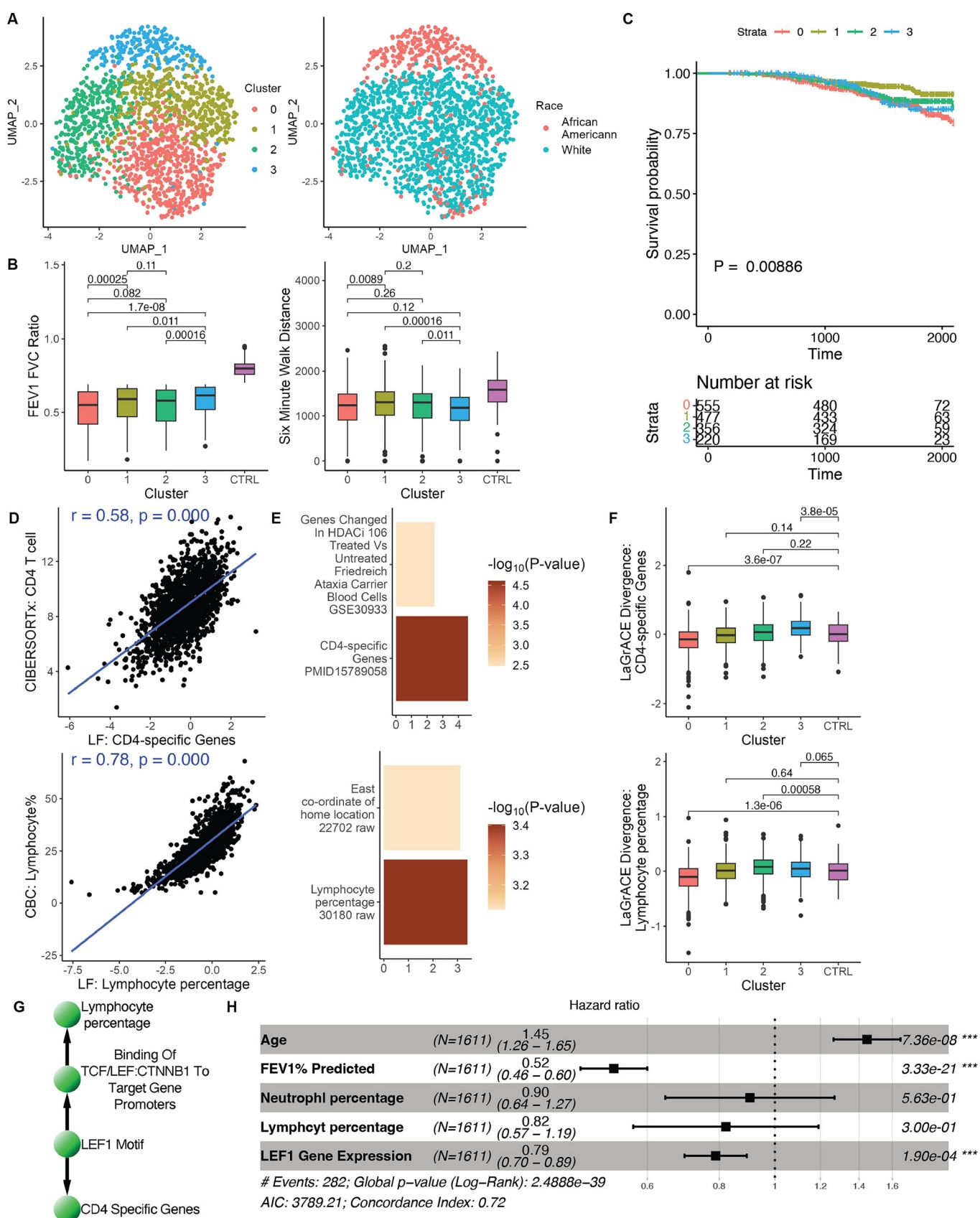

◀ **Figure 5. LaGrACE identified novel clusters on COPD dataset.**

(A) UMAP visualization of COPD patients colored by LaGrACE clusters and race. (B) Boxplot of FEV1 FVC ratio and 6-min walk distance (*P* values: Wilcoxon rank-sum test). Patients were stratified into a Reference Group ($n = 139$) and LaGrACE clusters (Cluster 0: $n = 556$; Cluster 1: $n = 478$; Cluster 2: $n = 357$; Cluster 3: $n = 220$). (C) Kaplan–Meier survival curve of COPD patients by LaGrACE clusters (*P* values: log-rank test). (D) Top, the Pearson correlation between LF: lymphocyte percentage and lymphocyte percentage from complete blood count (CBC). Bottom, the correlation between LF: CD4-specific genes and cell-type fraction sum of inferred CD4 T-cell subpopulations using CIBERSORTx. (E) Gene set enrichment visualization barplots of LF: lymphocyte percentage (UK Biobank GWAS) and LF: CD4-specific genes (HDSigDB Human) (*P* values: Fisher's Exact Test). (F) Boxplot of LaGrACE score of LF: lymphocyte percentage and LF: CD4-specific genes (*p* values: Wilcoxon rank-sum test). Patients were stratified into a Reference Group ($n = 139$) and LaGrACE clusters (Cluster 0: $n = 556$; Cluster 1: $n = 478$; Cluster 2: $n = 357$; Cluster 3: $n = 220$). (G) Subgraph of reference network. Nodes are annotated based on gene set enrichment analysis. (H) Forest plot summarizing the result of a multivariate cox proportional hazards regression model ($n = 1611$; excluding samples from the Reference Group). Error bars represent 95% confidence intervals for the hazard ratios. Box plots indicate the median (50th percentile), the lower (25th percentile) and upper (75th percentile) quartiles. The whiskers extend to the minimum and maximum values within 1.5 times the interquartile range (IQR) below the 25th percentile and above the 75th percentile, respectively. Data points beyond 1.5×IQR and less than 3×IQR are shown as whiskers, while those beyond 3×IQR are considered extreme outliers. Source data are available online for this figure.

regression analysis that adjusted for other clinical factors (Fig. 5H). Also, LEF1 gene expression was found to be prognostic for both white and African American patients (Appendix Fig. S13C).

Further investigation into the association between LEF1 and COPD disease progression revealed that LEF1 gene expression correlated significantly with age, FEV1%predicted, and FEV1/FVC ratio (Table 3). Importantly, the positive correlation between LEF1 and both spirometric measures (FEV1% predicted, FEV1/FVC) persisted after controlling for age. To validate the association between LEF1 and pulmonary function, blood-derived gene expression microarray data from the ECLIPSE study were employed. This dataset captured expression levels of two LEF1 gene regions: the full-length LEF1 and a partial region. Only the full-length LEF1 exhibited a positive correlation with FEV1% predicted and FEV1/FVC ratio (Table 3, ECLIPSE cohort). This association was not observed for the partial region lacking the high-mobility group domain(Kobielak, Kobielak, and Trzeciak 2001) (Table 3). Consistently, CD4 and CD8 T cells from COPD lung tissue, mirroring the transcriptional state of CD4 memory T cells from blood samples of COPD patients, exhibited lower LEF1 regulon scores (Appendix Fig. S14).

## Discussion

In this paper, we introduce LaGrACE, a novel method to quantify sample-level gene program dysregulation in observational biomedical datasets. Unlike other methods, LaGrACE employs a causal inference framework to estimate how gene program regulation varies in each noncontrol sample (e.g., disease or treatment) using control samples as a reference. Our empirical results demonstrate that contextual regulatory networks inferred from control samples help us to better depict of gene program dysregulation caused by disease or treatment and allow for grouping of samples with similar dysregulation patterns, facilitating biological discovery.

In simulation studies, LaGrACE demonstrates robust performance in characterizing alteration of the regulatory network across varying reference sample sizes and dimensionalities. It outperforms existing single-cell gene program methods without *prior knowledge* and effectively discerns between drug treatment signals across different doses and exposure times at a single-cell resolution. In addition, LaGrACE effectively identifies biologically significant and prognostic clusters in breast cancer data (METABRIC). The stability of these identified clusters was affirmed through validation

in the SCAN-B cohort, illustrating our method's reliability in patient stratification. Notably, these clusters provide deeper prognostic insight than molecular subtypes and clusters deduced from *prior knowledge*-based methods. Upon constructing cluster-specific networks, we observed significant associations between disease progression in ER-positive breast cancer patients and key signaling pathways, including the apical junction system and E2F transcription factor. These findings suggest that LaGrACE is a potent tool for elucidating the underlying mechanisms of diseases characterized by complex and heterogeneous pathophysiological profiles.

Furthermore, we also identify a novel association between the full-length LEF1 isoform and COPD molecular mechanisms and mortality. This association suggests that differential expression of LEF1 isoforms in COPD patients may perturb canonical WNT signaling and lymphopoiesis, consequently altering immune response (Feder et al, 2020; Cadigan and Waterman, 2012; Söderholm and Cantù, 2021; Hoppler and Kavanagh, 2007). In addition, the phenomenon of cellular senescence is known to correlate with several hallmarks of COPD pathogenesis (Araya and Kuwano, 2022). Declines in LEF1 expression levels and the ratio of full-length to truncated LEF1 isoforms with aging have been implicated in contributing to cellular senescence, thereby exacerbating COPD pathogenesis (Elyahu et al, 2019; Jia et al, 2023).

In the future, LaGrACE's ability to accurately characterize regulatory network dysregulation and group the samples based on similarity of dysregulation patterns positions it as a powerful tool for advancing personalized medicine. By identifying distinct gene programs across patient subgroups, LaGrACE can help clinicians identify individuals more likely to benefit from specific therapies or targeted interventions. For instance, in breast cancer, the subgroup-specific gene program dysregulation identified by LaGrACE can be used to guide the selection of targeted therapies tailored to each patient's unique molecular profile, ultimately improving treatment outcomes, and promoting personalized care.

Despite its strengths, LaGrACE has some limitations. Its underlying assumptions about linearity between observed variables ($X$) and latent factors ($Z$), as well as between clinical features ($Y$) and $Z$, may not always align with the biological nature of the data. Other limitations include the assumption of *acyclicity* between latent factors and the *Gaussianity* assumption for $X$ and $Y$. The assumption of acyclicity, in particular, may not fully capture feedback mechanisms that are often present in complex biological systems. In cases where strong feedback mechanisms are expected,

**Table 2.   Clinical characteristics of COPD participants vary across clusters.**

| Variable | Reference group | Cluster 0 | Cluster 1 | Cluster 2 | Cluster 3 | # missing | P value |
|---|---|---|---|---|---|---|---|
| Participants, n | 139 | 556 | 478 | 357 | 220 | 0 | |
| Age at Phase 2 visit, yr, mean (SD) | 64.527 (7.544) | 69.182 (8.034) | 68.867 (8.375) | 69.832 (8.045) | 62.075 (7.413) | 0 | |
| Gender, female, n (%) | 78 (56.12) | 224 (40.29) | 230 (48.12) | 136 (38.1) | 110 (50) | 0 | |
| Race, African American, n (%) | 29 (20.86) | 60 (10.79) | 31 (6.49) | 28 (7.84) | 217 (98.64) | 0 | |
| Race, White, n (%) | 110 (79.14) | 527 (94.78) | 447 (93.51) | 329 (92.16) | 3 (1.36) | 0 | |
| Gold Stage 0, n | 139 | NA | NA | NA | NA | 0 | |
| Gold Stage 1, n | NA | 120 | 113 | 93 | 49 | 0 | |
| Gold Stage 2, n | NA | 226 | 224 | 149 | 117 | 0 | |
| Gold Stage 3, n | NA | 146 | 99 | 83 | 41 | 0 | |
| Gold Stage 4, n | NA | 64 | 42 | 32 | 13 | 0 | |
| **Complete blood count measurement** | | | | | | | |
| Lymphocyte percentage, mean (SD) | 32.266 (9.522) | 22.016 (7.14) | 28.912 (8.277) | 26.764 (8.072) | 36.914 (9.046) | 5 | 1.34E-89 |
| Neutrophil percentage, mean (SD) | 56.417 (9.616) | 66.959 (8.596) | 58.895 (8.66) | 61.138 (8.856) | 51.645 (10.072) | 5 | 1.24E-85 |
| Neutrophil, K/μL, mean (SD) | 3.792 (1.504) | 5.216 (1.833) | 4.404 (1.391) | 4.625 (1.523) | 3.327 (1.429) | 7 | 8.31E-48 |
| Lymphocytes, K/μL, mean (SD) | 2.062 (0.661) | 1.636 (0.578) | 2.121 (0.774) | 1.985 (0.775) | 2.256 (0.718) | 7 | 5.88E-38 |
| Neutrophil-to-lymphocyte ratio percentage, mean (SD) | 2.034 (1.184) | 3.665 (2.312) | 2.345 (1.267) | 2.679 (1.583) | 1.622 (1.473) | 7 | 1.30E-90 |
| **CIBERSORTx inferred cell-type fraction** | | | | | | | |
| Neutrophils percentage, mean (SD) | 11.132 (4.05) | 14.955 (4.689) | 10.968 (3.337) | 13.106 (4.079) | 9.207 (3.457) | 0 | 6.83E-75 |
| CD4 memory resting T-cell percentage, mean (SD) | 4.126 (1.16) | 3.45 (1.184) | 4.116 (1.093) | 3.578 (1.165) | 4.81 (1.237) | 0 | 5.79E-48 |
| CD4 naive T-cell percentage, mean (SD) | 4.661 (1.56) | 3.637 (1.136) | 4.26 (1.331) | 4.493 (1.343) | 4.626 (1.386) | 0 | 1.37E-29 |
| Plasma cell percentage, mean (SD) | 0.172 (0.168) | 0.283 (0.213) | 0.178 (0.147) | 0.248 (0.182) | 0.163 (0.143) | 0 | 1.77E-26 |
| **Symptoms** | | | | | | | |
| MMRC dyspnea score, mean (SD) | 0.626 (1.031) | 1.766 (1.459) | 1.454 (1.389) | 1.529 (1.468) | 1.95 (1.469) | 0 | 2.66E-05 |
| CAT score, mean (SD) | 7.36 (5.992) | 14.959 (8.842) | 13.134 (8.526) | 13.958 (8.351) | 16.086 (9.092) | 0 | 7.58E-05 |
| SGRQ score: active, mean (SD) | 21.225 (23.326) | 47.811 (28.956) | 41.837 (28.896) | 43.972 (29.521) | 49.734 (30.07) | 0 | 9.48E-04 |
| Six-min walk distance, ft, mean (SD) | 1530.109 (408.52) | 1174.192 (465.711) | 1258.888 (425.263) | 1211.714 (439.033) | 1132.23 (438.257) | 39 | 1.38E-03 |
| SGRQ score: total, mean (SD) | 12.548 (13.811) | 32.415 (21.92) | 28.492 (21.018) | 30.077 (21.368) | 34.1 (22.435) | 0 | 4.14E-03 |
| **Spirometry** | | | | | | | |
| Total lung capacity, Thirona, liters, mean (SD) | 5.203 (1.214) | 6.108 (1.41) | 6.049 (1.478) | 6.251 (1.46) | 5.014 (1.231) | 153 | 4.73E-22 |
| FRC/TLC ratio, Thirona, mean (SD) | 0.513 (0.084) | 0.645 (0.111) | 0.625 (0.109) | 0.627 (0.107) | 0.686 (0.109) | 256 | 5.60E-09 |
| FEV1/FVC ratio, post-BD, mean (SD) | 0.798 (0.053) | 0.526 (0.129) | 0.554 (0.122) | 0.541 (0.125) | 0.583 (0.098) | 1 | 1.32E-07 |
| DLCO percent predicted, GLI, mean (SD) | 92.289 (16.349) | 64.716 (22.658) | 70.411 (22.755) | 66.901 (21.915) | 62.546 (19.214) | 258 | 4.79E-05 |
| BODE score, mean (SD) | 0.511 (1.03) | 2.871 (2.594) | 2.316 (2.348) | 2.463 (2.405) | 2.889 (2.312) | 85 | 9.72E-04 |
| **Smoking status** | | | | | | | |
| Smoked in the past month, yes, n (%) | 0 (0) | 160 (28.78) | 140 (29.29) | 107 (30.06) | 140 (63.64) | 1 | 1.48E-21 |
| Smoked over the past 5 years, yes, n (%) | 22 (15.83) | 206 (37.18) | 195 (40.88) | 138 (38.87) | 164 (74.55) | 5 | 1.99E-21 |

P values were calculated using a Kruskal–Wallis test for continuous and ordinal variables, and a Chi-squared test for discrete variables, to assess if there are differences in the distribution of variables among clusters.

BD bronchodilator, BODE body mass index, airflow obstruction, dyspnea, and exercise capacity, CAT COPD Assessment Test, DLCO diffusing capacity for carbon monoxide, FEV1 forced expiratory volume in 1-s, FRC functional residual capacity, FVC forced vital capacity, MMRC Modified Medical Research Council Dyspnea Scale, SGRQ St. George's Respiratory Questionnaire, TLC total lung capacity.

Table 3.   Correlation analysis between LEF1 gene expression and clinical variables.

| Correlation analysis | Variable | PCC (r) | P value | SRCC (rho) | P value |
|---|---|---|---|---|---|
| LaGrACE divergence score of LF: lymphocyte percentage | FEV1 FVC ratio ($n = 1749$) | **0.12** | **1.34E-06** | **0.10** | **6.05E-05** |
| | FEV1% predicted ($n = 1750$) | **0.10** | **1.22E-04** | **0.08** | **1.18E-03** |
| LaGrACE divergence score of LF: CD4-specific genes | FEV1 FVC ratio ($n = 1749$) | **0.20** | **8.68E-16** | **0.19** | **7.08E-14** |
| | FEV1% predicted ($n = 1750$) | **0.11** | **1.41E-05** | **0.11** | **1.52E-05** |
| COPDgene: LEF1 gene expression | Age ($n = 1750$) | **−0.31** | **7.81E-40** | **−0.32** | **3.20E-43** |
| | FEV1 FVC ratio ($n = 1749$) | **0.20** | **1.17E-16** | **0.19** | **3.19E-15** |
| | FEV1% predicted ($n = 1750$) | **0.18** | **7.10E-14** | **0.16** | **9.53E-12** |
| ECLIPSE: LEF1 partial region (AF294627) | Age ($n = 229$) | **−0.17** | **8.22E-03** | **−0.24** | **2.72E-04** |
| | FEV1 FVC ratio ($n = 229$) | 0.04 | 5.15E-01 | 0.09 | 1.96E-01 |
| | FEV1% predicted ($n = 229$) | 0.04 | 5.58E-01 | 0.07 | 2.69E-01 |
| ECLIPSE: LEF1 full length (AF288571) | Age ($n = 229$) | **−0.25** | **1.51E-04** | **−0.25** | **1.19E-04** |
| | FEV1 FVC ratio ($n = 229$) | **0.25** | **9.95E-05** | **0.27** | **4.66E-05** |
| | FEV1% predicted ($n = 229$) | **0.24** | **1.94E-04** | **0.25** | **1.25E-04** |
| Partial correlation analysis | Variable | PCC (r) | P value | SRCC (rho) | P value |
| COPDgene: LEF1 gene expression condition on AGE | FEV1 FVC ratio ($n = 1749$) | **0.16** | **4.54E-11** | **0.15** | **1.11E-09** |
| | FEV1% predicted ($n = 1750$) | **0.18** | **3.17E-14** | **0.16** | **1.55E-11** |
| ECLIPSE: LEF1 partial region (AF294627) condition on AGE | FEV1 FVC ratio ($n = 229$) | 0.00 | 9.70E-01 | 0.04 | 5.41E-01 |
| | FEV1% predicted ($n = 229$) | 0.00 | 9.92E-01 | 0.04 | 5.85E-01 |
| ECLIPSE: LEF1 full-length (AF288571) condition on AGE | FEV1 FVC ratio ($n = 229$) | **0.20** | **2.11E-03** | **0.23** | **5.33E-04** |
| | FEV1% predicted ($n = 229$) | **0.20** | **2.38E-03** | **0.22** | **8.24E-04** |

*PCC* Pearson correlation coefficient, *SRCC* Spearman's rank correlation coefficient.
Bold values are statistically significant.

alternative methods like the Fast Causal Inference (FCI) algorithm could be used to infer regulatory networks more effectively (Mooij and Claassen 2020). Nonetheless, the linearity assumption between $X$ and $Z$ is a reasonable approximation in many cases. This is also reflected on our results in real-life datasets (breast cancer, COPD, and the single-cell analysis). In the future, we plan to extend LaGrACE to non-Gaussian and non-linear models.

# Methods

**Reagents and tools table**

| Reagent/resource | Reference or source | Identifier or catalog number |
|---|---|---|
| **Experimental models** | | |
| MDA-MB-231 human breast cancer cell line | Dr. Gary Luker's Lab, University of Michigan | |
| SUM149 human breast cancer cell line | Dr. Gary Luker's Lab, University of Michigan | |
| SUM159 human breast cancer cell line | Dr. Gary Luker's Lab, University of Michigan | |
| **Recombinant DNA** | | |
| pEF1alpha-tdTomato Vector | Takara | 631975 |
| **Antibodies** | | |

| Reagent/resource | Reference or source | Identifier or catalog number |
|---|---|---|
| **Oligonucleotides and other sequence-based reagents** | | |
| **Chemicals, enzymes, and other reagents** | | |
| DMEM | Gibco | 11995 |
| F-12 | Gibco | 11765 |
| Fetal Bovine Serum (FBS) | Gibco | 16000 |
| GlutaMAX | Gibco | 35050 |
| Penicillin/Streptomycin | Gibco | 15070 |
| Trypsin-EDTA (0.05%) | Gibco | 25200 |
| Hydrocortisone | Sigma | H4001 |
| Insulin | Sigma | I6634 |
| Plasmocin | InvivoGen | ant-mpp |
| Xfect Transfection Reagent | Takara | 631317 |
| G418 | Takara | 631307 |
| Collagen Type I | BD Biosciences | 354236 |
| Thiostrepton | Cayman Chemical | 19200 |
| FDI-6 | MedChemExpress | HY-112721 |
| RCM-1 | MedChemExpress | HY-19979 |
| DMSO | Sigma or equivalent | |
| Acetic acid | Sigma or equivalent | |
| **Software** | | |

| Reagent/resource | Reference or source | Identifier or catalog number |
|---|---|---|
| MATLAB | MathWorks | |
| GraphPad Prism | GraphPad | v9.3 |
| **Other** | | |
| Custom single-cell microfluidic migration device | Zhou et al, 2023; Chen et al, 2019 | |

## LaGrACE algorithm

LaGrACE is an algorithm that evaluates gene program dysregulation through estimating pseudo-control outcome on a gene program or a clinical variable. LaGrACE comprises four main steps, which are summarized below and elaborated in the following sections.

(1) **Gene Program Discovery**: infer biologically meaningful latent factors (gene programs) from gene expression profiles of reference samples and then calculate factor matrix for treatment group using loading matrix inferred based on reference samples.
(2) **Reference Network Learning**: learn a Bayesian network encapsulated interactions among inferred gene programs and clinical variables of interest.
(3) **Pseudo Control Outcome Estimation**: estimate pseudo-control outcome to approximate pre-treatment state on a gene program or a clinical measurement of disease (or treated) samples by utilizing its Markov blanket derived from the reference network.
(4) **Divergence Estimation**: calculate the difference between the observed value and the pseudo-control outcome for treatment samples.

In most biomedical study, there are omics (gene expression) data and clinical variables related to a specific treatment (or disease). Each sample in the dataset is labeled to denote whether it has received this treatment. Assume that the treatment is well-defined, and feature interactions are invariant across reference samples, we denote an individual sample in treatment state by W = 1 and the reference one by W = 0.

As a notation, let $X \in R^{N_{genes} \times N_{samples}}$ represent gene expression matrix, and $Z \in R^{k \times N_{samples}}$ represent k gene programs inferred from gene expression profile X, where $k < N_{genes}$. The matrix $A \in R^{N_{genes} \times K}$ represents the membership matrix assigning observed gene expression features to $k$ latent factors. Since biomedical data usually include clinical measurements, we denote $U \in R^{N_{variables} \times N_{samples}}$ represent clinical variable matrix, where clinical variable may be either continuous or discrete.

The reference network, representing conditional independence relations among variables, delineates interactions among gene programs inferred from gene expression profiles and clinical variables. Let $V = [Z, U]$ denote random variables in the network, and $V_{ij}^0$ be value of feature i (could be gene program Z or clinical variable of interest U) in a sample j if the value was observed from an individual with a treatment label W = 0. For an individual j from treatment group, $V_{ij}^1$ represents observed value of feature i, but $V_{ij}^0$ is unknown. We infer pseudo-control outcome $\hat{V}_{ij}^0$ to approximate $V_{ij}^0$. We focus on understanding its impact without the complication of multiple treatments or interference effects.

## Gene program discovery

The (soft) clustering approach LOVE (Bing et al, 2019), a robust and scalable distribution-free latent model-based statistical method, is capable of inferring biologically meaningful and non-orthogonal latent factors. The latent factor model is formularized as

$$X = AZ + E$$

The model identifiability requires a mild condition that, each latent factor needs to have at least two primary variables assigned uniquely to it. Unlike other factorization models, this method allows a slight departure from independence among latent factors that are inferred from observed gene expression features. Hyperparameters (delta) were selected using Akaike Information Criterion (AIC) method, where delta parameter controls the number of latent factors.

## Reference network learning

Once gene programs are identified, interactions among those gene programs and clinical variables of interest are modeled using the Fast Greedy Equivalent Search (FGES) algorithm (Ramsey et al, 2017) on reference samples. FGES constructs a partially Directed Acyclic Graph (pDAG), which contains both directed and undirected edges and can be used to represent a Markov equivalence class of the data-generating directed acyclic graph (DAG) *G*, over all input variables. FGES holds the Causal Markov assumption, stating that each variable in a graphical model is conditionally independent of every other variable (that is not affect by the variable), conditional on the variable's direct causes. FGES starts its search with an empty graph, then adds a single edge increasing DG score. Finally, it removes single edges while the score increases.

To accommodate the potential for discrete features within clinical data, FEGS was performed by maximizing degenerate Gaussian (DG) score (Andrews et al, 2019). Let $V = \left(V_1, \cdots, V_p\right)$ be a set of p random variables with n instances, where $V_i$ may be either continuous (**C**) or discrete (**D**). The DG score relies on standard transformation (One-hot Encoding) of variables:

$$T_i = \begin{cases} V_i \, if \, i \in C \\ l_1(V_i), \dots, l_{k-1}(V_i) \, if \, i \in D \end{cases}$$

Where $l_k(V_i)$ is an indicator function such that $l_k(V_i) = 1$ if $V_i = k$, otherwise $l_k(V_i) = 0$ for $V_i \in D$. The One-hot Encoding transformation embeds the discrete variables into a degenerate distribution. Thus, the transformed data is further treated as jointly Gaussian. The DG score is the sum of local components (its nodes and their parents):

$$S(G, T) = \sum_{i=1}^{p} s\left(T_i, T_{Pa_i^G}\right)$$

*where* $T_{Pa_i^G}$ represents the set of parents of $T_i$, and

$$s\left(T_i, T_{Pa_i^G}\right) = l\left(\hat{\theta}_{mle} | T_{\{i\} \cup Pa_i^g}\right) - l\left(\hat{\theta}_{mle} | T_{Pa_i^G}\right) \\ - \frac{PD}{2} |T_i| \left|T_{Pa_i^G}\right| \log(n)$$

PD is a penalty value (penalty discount) that controls the sparsity of graph. $l(\hat{\theta}_{mle} | T_{\{i\} \cup Pa_i^g})$ and $l(\hat{\theta}_{mle} | T_{Pa_i^G})$ are Gaussian log-

likelihood of the transformed data T given the causal DAG $G$ and its parameters ($\theta$).

## Pseudo-control outcome estimation

The pseudo-control outcome approach is designed to approximate pre-treatment state on a gene program or a clinical measurement by utilizing its Markov blanket based on the reference network. The Markov blanket of a variable $V_i$, $MB(V_i)$, consists of the parents, children, and spouses, which are connected by adjacent directed or undirected edges under the causal Markov assumption. The variable of interest is considered independent of all other variables, conditioned on its Markov blanket. Once the reference network is estimated, all discrete variables are transformed into binary indicator variables, representing categories of the original discrete variable. A set of Markov blanket-based regression models are then built for each variable in the reference network using reference samples.

$$V_{i\cdot}{}^0 = \sum_{V_k{}^0 \in MB(V_i{}^0)} \beta_{k,i}{}^0 V_{k\cdot}{}^0 + \beta_{0,i}{}^0 + \varepsilon$$

where $V_{i\cdot}$ represents variable $i$ across reference samples.

Given the optimized coefficients, we predict pseudo-control outcome of a variable $V_i$ for each sample j from treatment group.

$$\hat{V}_{ij}^0 = \sum_{V_k{}^0 \in MB(V_i{}^0)} \beta_{k,i}{}^0 V_{kj}{}^1 + \beta_{0,i}{}^0 + \varepsilon$$

## Divergence estimation

Causal reasoning involves comparing the outcome when a treatment is administrated ($V^1$) against when it is not ($V^0$). For simplicity, we have omitted the subscripts i and j, which denote the same variable and sample. A causal effect on the outcome is inferred if these outcomes differ ($V^1 \neq V^0$). Individual causal effects are defined as contrasts between counterfactual outcomes, but for each individual, only one of these outcomes can be observed. To approximate the unobserved counterfactual outcome, we estimate a pseudo-control outcome based on our understanding of the gene program regulatory network. The divergence score, which quantifies the dysregulation of a variable i within reference network or the causal effect induced by disease or treatment, is calculated by comparing the pseudo-control outcome with observed feature value of variable $V_i$ in sample j from the treatment group.

$$\text{Divergence Score}: V_{ij}^1 - \hat{V}_{ij}^0 = V_{ij}^1 - \left( \sum_{k \in MB(V_i)} \beta_{k,i}{}^0 V_{kj}{}^1 + \beta_{0,i}{}^0 \right)$$

Specifically, gene program divergence score can be defined as $Z_{ij}^1 - \hat{Z}_{ij}^0$.

The divergence score across all variables in the reference network can also be further coupled with $k$-fold cross-validation on the reference samples. This approach assists in selecting the sparsity penalty value of FGES that gives the most informative reference network.

## Sample clustering

Using the *Seurat* package (Hao et al, 2021), we conducted single-sample clustering, comparing clusters based on LaGrACE divergence score and other methods. Firstly, we applied linear dimensional reduction, projecting samples into principal component space. The optimal number of principal components was determined by the JackStraw method or variation between consecutive PCs. The *clustree* package was used to determine appropriate clustering resolution for stable clusters (Zappia and Oshlack, 2018). A graph-based clustering was performed with constructed KNN graph based on the Euclidean distance in Principal Component Analysis (PCA) space and then optimized the standard modularity function. Finally, non-linear dimensional reduction (UMAP) was performed to visualize the dataset in a low-dimensional space. For subtyping on synthetic datasets, we used 20 principal components for clustering (to match the *Seurat* default value). In synthetic dataset subtyping, we used 20 principal components with a minimal clustering resolution, ensuring at least three clusters and each cluster accounting for less than 60% of total perturbed samples.

## Synthetic dataset evaluation

To assess LaGrACE's efficacy in identifying subtypes from mixed-type data, comprising both continuous and discrete variables, we utilized the Lee and Hastie (LH) model (Lee and Hastie, 2015). This model delineates the joint distribution over datasets with mixed variables. We first simulated a reference network with a vertex degree of 2 and then created three perturbed networks through random edge modifications (15% of the total edges, including both deletion and addition). Each network contained 50 continuous and 25 discrete variables, the latter with four categorical levels. Datasets of 1000 samples were simulated for each network based on the LH model, with continuous variables drawn from $\mathcal{N}(\mu, \sigma)$, where $\mu \sim U(-1, 1)$ and $\sigma \sim U(0.5, 1)$. Edge weights, $W_{X1 \times 2}$, were assigned from a uniform distribution $U(0.5, 1.5)$. We assumed linear interactions for discrete-continuous and discrete-discrete edges. For a comprehensive evaluation, we generated 10 datasets, each with 1000 samples from the reference network and 3000 from the three perturbed networks.

### Biologically inspired synthetic high-dimensional dataset

In the analysis of high-dimensional and collinear omics datasets, such as transcriptomics profiles, we modeled gene expression data using LH data simulated from one reference network and three perturbed networks. This model incorporated 50 continuous variables representing latent factors (Z) (e.g., transcriptional factor (TF) protein activities), and 25 discrete variables, analogous to clinical features. We then simulated 2500 observed variables (X), representing "gene expression", from the 50 latent factors (Z) and their loadings (A) using

$$X = AZ + E$$

where $E \sim \mathcal{N}(0, 1)$ denotes independent noise. Each Z factor directly influenced ~50 observed features, and A was simulated with weights from a Gaussian distribution $\mathcal{N}(0, 1)$. Only 1% of the values

in the loading matrix with the largest absolute values were retained, while the rest were set to zero. We generated 10 datasets, each comprising 1000 "normal" samples from the reference network and 3000 "treatment" samples from the three perturbed networks.

## Application to breast cancer and COPD

### Cancer datasets and preprocessing
We applied LaGrACE to two cancer datasets containing RNA-seq and clinical data: the METABRIC Dataset (Rueda et al, 2019) and the SCAN-B Dataset (Saal et al, 2015). METABRIC data were downloaded from cBioPortal and METABRIC study (Rueda et al, 2019), and SCAN-B data from NCBI GEO (GSE96058). METAB-RIC dataset was split into several molecular subtypes: basal, claudin-low, HER2 +, luminal A, luminal B, and normal breast-like. SCAN-B dataset was comprised of basal group, HER2+ group, luminal A group, and luminal B group. The normal-like group of primary specimens is usually composed of actual normal breast samples, but it also includes a small number of tumor samples. Several studies questioned the existence of this subtype and suspected that the tumor samples in the normal breast-like group may come from (adjacent) normal breast tissue. Thus, the normal-like samples are excluded from downstream analysis (Bernard et al, 2009; Prat et al, 2010). Besides, samples from subjects that died of other causes were also excluded.

A common set of 16917 genes was used for both datasets, with Z-score normalization to mitigate batch effects. Six clinical features (breast carcinoma estrogen receptor status, breast carcinoma progesterone receptor status, her2 immunohistochemistry receptor status, age at diagnosis, lymph nodes examined positive, and tumor size) were included for mixed-type data analysis, and samples with missing values were excluded.

### COPD datasets and preprocessing
LaGrACE analysis was conducted based on clinical and blood-derived biological features (white blood cell differential percentages, whole blood gene expression) from COPD Genetic Epidemiology (COPDGene) Study (Ghosh et al, 2022). The ECLIPSE study (Vestbo et al, 2008) and scRNA-seq data of human lung (Adams et al, 2020) was used for external validation of our findings.

### Data analysis
Kaplan–Meier survival curves and recurrence events were analyzed using *survival* and *survminer* packages. Time-dependent AUC curve plots were generated using timeROC package. Discriminatory power of identified clusters was accessed by Harrell's concordance index (C-index), Gönen & Heller's concordance probability estimate (CPE), and time-dependent ROC curve. Differences in clinical variables among clusters were evaluated using Chi-squared tests for categorical variables and Kruskal–Wallis or Wilcoxon rank-sum tests for continuous/ordinal variables. The FCI-stable algorithm was performed using the *rCausalMGM* package (Lovelace, Benos, in preparation).

We calculated the epithelial cell differentiation score (Prat et al, 2010), proliferation index (Nielsen et al, 2010), and immune cytolytic activity (Rooney et al, 2015) on all tumor samples. The differentiation score, as a measure of mammary development, was computed using a centroid-based predictor with information from

approximately 20,000 gene expression features. The proliferation signature index was derived from 11 genes (BIRC5, CCNB1, CDC20, NUF2, CEP55, NDC80, MKI67, PTTG1, RRM2, TYMS, and UBE2C), and immune cytolytic activity from the geometric mean of GZMA and PRF1 expression.

Pathway enrichment was assessed through ssGSEA (Barbie et al, 2009), MSigBD genesets and *GSVA* package. Transcriptional factor regulon scores were inferred using *decoupleR* (Badia-i-Mompel et al, 2022). Cell-type fractions in bulk transcriptome profiles were deconvoluted using CIBERSORTx (Newman et al, 2019) based on immune population gene signature matrix (LM22) and breast cancer single-cell profiles (Wu et al, 2021). Enrichr (Xie et al, 2021) was employed to find overlaps between genes assigned to latent factor of interest and precompiled ontologies.

COPD module scores on T lymphocyte from human lung were computed based on differentially expressed genes associated with COPD, identified in memory CD4 T cells from blood using *AddModuleScore* function from *Seurat* package. To control for effect of aging, the comparison is restricted to samples from individuals aged 55–70 years.

### Assignment of cluster membership to new samples
To assign cluster labels to the validation dataset based on LaGrACE features, we trained a random forest classifier with LaGrACE features from the METABRIC dataset using *randomForest* package. Parameters were tuned using tenfold cross-validation with the *caret* and *mlbench* package. To minimize batch effects, we calibrated the model learned from the METABRIC dataset with basal samples (reference group) from the SCAN-B dataset, then calculated LaGrACE features for samples from the SCAN-B dataset.

## Application to a single-cell drug response dataset

To evaluate LaGrACE suitability for single-cell data, we used two public scRNA-seq datasets: GSE117089, featuring a time series of dexamethasone treatment on A549 human lung adenocarcinoma cells (Cao et al, 2018); and GSE139944, containing data on A549 cells treated with increasing doses of small molecules (BMS-345541, nutlin-3A, SAHA, or DMSO vehicle control) for 24 h (Srivatsan et al, 2020).

### Data analysis
Both scRNA-seq datasets were processed using the *Seurat* pipeline. The deep generative model scVI (Lopez et al, 2018) was applied to impute the top 3000 highly variable genes from each dataset. In the LaGrACE analysis, the vehicle group was selected as the reference group.

## Method comparison

### Pathway activity quantification method
Pathifier and ssGSEA were used for bulk RNA-seq datasets. We used the MSigBD C2 gene set collection for ssGSEA. For Pathifier, we applied KEGG pathway information from *KEGGREST* and the *pathifier* package for pathway deregulation scoring.

### Single-cell gene program score method
To run Spectra, expression data was normalized using sran. We used the default parameters with 151 default Spectra global

genesets. For expiMap, we used default parameters with Reactome genesets used in expiMap tutorial.

## Evaluation metrics

### Clustering evaluation
Clustering performance was assessed using the Adjusted Rand Index (ARI) and Adjusted Mutual Information (AMI) via the *aricode* package.

### Single-cell neighborhood evaluation
For performance evaluation on scRNA-seq datasets without relying on clustering, Milo (Dann et al, 2022) was used to model cell states as neighborhoods on a KNN graph. This graph was constructed using PCA based on input features, such as RNA or LaGrACE features. Cell-type purity for each neighborhood, defining it as the frequency of the most prevalent cell type, was calculated utilizing dose information or treatment time (see Appendix Text S1 for details).

## Scalability and runtime

The computation of LaGrACE divergence scores involves four main steps, each with distinct computational requirements. First, Gene Program Discovery is performed using the LOVE algorithm to identify latent factors corresponding to gene programs, with a time complexity of $O(p^2)$, where p represents the number of observed features. Second, Reference Network Learning employs the FGES algorithm, which also has a time complexity of $O(p^2)$, where p denotes the number of input variables. Third, Pseudo Control Outcome Estimation fits a linear regression model, with a time complexity of $O(n \cdot p2 + p3)$, where $n$ is the number of observations and p is the number of variables (including the variable of interest and its Markov blanket). Finally, Divergence Estimation computes element-wise differences, requiring a time complexity of $O(n)$, where n is the number of observations.

To evaluate LaGrACE's performance, we tested it on two datasets using a 3 GHz 6-Core Intel Core i5 iMac: scRNA-seq data from 5530 A549 cells treated with SAHA (including 1005 reference samples) with 3000 highly variable genes and simulated high-dimensional data with 4000 samples (1000 reference samples) containing 2500 continuous and 25 discrete variables (Appendix Table S5).

## Cell culture

We cultured MDA-MB-231 human epithelial cells in Dulbecco's modified Eagle medium (DMEM, Gibco 11995) supplemented with 10% fetal bovine serum (FBS, Gibco 16000), 1% GlutaMax (Gibco 35050), 1% penicillin/streptomycin (pen/strep, Gibco 15070), and 0.1% of plasmocin (InvivoGen ant-mpp). We cultured SUM149 and SUM159 human breast cancer cells in F-12 (Gibco 11765) media supplemented with 5% FBS (Gibco 16000), 1% pen/strep (Gibco 15070), 1% GlutaMax (Gibco 35050), 1 µg mL$^{-1}$ hydrocortisone (Sigma H4001), and 5 µg mL$^{-1}$ insulin (Sigma I6634), and 0.1% of plasmocin (InvivoGen, ant-mpp). These breast cancer cells were obtained from Dr. Gary Luker's lab at the University of Michigan. We maintained all cells at 37 °C in a humidified incubator with 5% $CO_2$. All the cells were cultured and passaged when the cells reached over 80% confluency in the dish.

Cell line authentication was guided by the recommendations of the International Cell Line Authentication Committee (ICLAC). All cell lines were cultured with mycoplasma antibiotics Plasmocin (Purchased from InvivoGen). Cell lines were routinely examined for mycoplasma contamination by sensitive PCR assays, and contaminated cells were discarded.

No blinding was done.

## Cell transfection

We transfected breast cancer cells using Xfect™ Transfection Reagent (Takara 631317) with 5 µg of pEF1alpha-tdTomato Vector plasmid (Takara 631975). The cells labeled with red fluorescent proteins can be tracked for migration. The transfected cells were selected using G418 (Takara 631307) treatment and sorted by flow cytometry for red fluorescence.

## High-throughput single-cell migration assay by microfluidics

The cell migration assays were performed using a microfluidic single-cell migration platform modified from our previous work (Zhou et al, 2023; Chen et al, 2019). The migration channel is designed to be 5 µm in height and 10 µm in width. The device was primed by a collagen solution (1.45 mL Collagen (Collagen Type 1, 354236, BD Biosciences) and 0.1 mL acetic acid in 50 mL DI Water) to enhance cell adhesion. Right before the migration experiment, the devices were rinsed by cell culture media. Breast cancer Cells were trypsinized, centrifuged, and resuspended to a concentration of $4 \times 10^5$ cells/mL for loading onto the devices. The devices were incubated for 30 min to enhance cell adhesion. FOXM1 inhibitors (Thiostrepton, 10 µM (Cat#: 19200), FDI-6, 10 µM (HY-112721), RCM-1, 10 µM (HY-19979)) were used to inhibit cell motility.

The microfluidic devices were incubated for 24 h, and the migration distance was measured based on the final cell position. The images were analyzed by a custom MATLAB code (Zhou et al, 2023). Cells were identified based on their fluorescence, and debris was ignored by their small size. In this work, two independent replicates (~200 migration channels) were performed. The Violin graphs were plotted using Prism (v.9.3).

As cell migration results do not follow normal distribution, a non-parametric Mann–Whitney $U$ test was used for comparisons of cell motility with a significance level of 0.05 considered statistically significant.

The step-by-step protocol for the migration assay follows.

## Step-by-step protocol of the migration assay

### Compound preparation

1. Dissolve each compound (Thiostrepton (Cayman: 19200), FDI-6 (Medchemexpress, HY-112721), RCM-1 (Medchemexpress, HY-19979)) to a concentration of 10 mM in either DMSO or PBS, following the vendor's instructions.
2. Prepare serial dilutions of the compound solutions.
3. Adjust the final working concentration to 10 µM for experiments.
4. Use 0.1% DMSO treatment as the control.

### Cell preparation

5. Harvest cells from culture plates using 0.05% Trypsin/EDTA (Gibco, 25200).
6. Centrifuge the cells at 1000 rpm for 4 min.
7. Resuspend the cell pellet in culture media to achieve a concentration of $4.17 \times 10^5$ cells/mL (equivalent to $2.5 \times 10^4$ cells per well).

### Cell Seeding in the migration device

8. Pipette 60 μL of the prepared cell suspension into the upper wells of the migration device.
9. Pipette 20 μL of culture media into the lower wells.
10. Ensure trypsinized rounded cells (diameter: 10–15 μm) are initially trapped at the entrance of migration channels (height: 5 μm).
11. Maintain the device under static flow conditions in an incubator for 30 min to allow cell adhesion.
12. Visually confirm cell adhesion morphology before proceeding to the next step.

### Compound treatment and chemotactic gradient setup

13. Aspirate the remaining cell suspension from the upper wells.
14. Replace with 60 μL of serum-free culture media containing the treatment compounds (10 μM).
15. Add 60 μL of serum culture media with treatment compounds (10 μM) to the lower wells to induce chemotactic migration.
16. Allow diffusion to establish a linear concentration gradient of the chemoattractant along the migration channel.

### Migration assay and analysis

17. Place the migration device in a cell culture incubator.
18. Incubate for 24 h without media replenishment.
19. Measure the migration distance based on the final cell frontier (the farthest migrating cell) in each migration channel.

## Data availability

The code used in this study is available in the following database: LaGrACE R Package and analysis code for synthetic and real-world data: GitHub (https://github.com/benoslab/LaGrACE).

The source data of this paper are collected in the following database record: biostudies:S-SCDT-10_1038-S44320-025-00115-3.

## Peer review information

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

## Acknowledgements

We would like to thank Prof. Kellie Smith, Johns Hopkins University for providing meta-data for the single-cell lung cancer dataset we analyzed in this paper. This work was partly supported by the following grants from the National Institutes of Health (NIH): R01HL178032, R01HL157879, R01HL127349, R01AA028436 (PVB); P30CA047904, P50CA272218, R35GM150509 (Y-CC). The COPDGene® study (NCT00608764) was supported by NHLBI U01HL089897, U01HL089856, and by the COPD Foundation through contributions made to an Industry Advisory Committee that has included AstraZeneca, Bayer Pharmaceuticals, Boehringer Ingelheim, Genentech, GlaxoSmithKline, Novartis, Pfizer, and Sunovion.

## Author contributions

**Minxue Jia**: Conceptualization; Data curation; Software; Formal analysis; Validation; Investigation; Visualization; Methodology; Writing—original draft; Writing—review and editing. **Haiyi Mao**: Conceptualization; Formal analysis; Validation; Investigation; Visualization; Methodology; Writing—original draft; Writing—review and editing. **Mengli Zhou**: Data curation; Formal analysis; Investigation; Visualization; Writing—original draft. **Yu-Chih Chen**: Data curation; Formal analysis; Funding acquisition; Investigation; Visualization; Writing—original draft. **Panayiotis V Benos**: Conceptualization; Data curation; Software; Formal analysis; Supervision; Funding acquisition; Validation; Investigation; Visualization; Methodology; Writing—original draft; Project administration; Writing—review and editing.

Source data underlying figure panels in this paper may have individual authorship assigned. Where available, figure panel/source data authorship is listed in the following database record: biostudies:S-SCDT-10_1038-S44320-025-00115-3.

## Disclosure and competing interests statement

The authors declare no competing interests.

