## [Peer Review File · Molecular Systems Biology]

LaGrACE: Estimating gene program dysregulation with latent regulatory networks

Minxue Jia, Haiyi Mao, Mengli Zhou, Yu-Chih Chen, and Panayiotis Benos

Corresponding author(s): Panayiotis Benos (pbenos@ufl.edu)

Review Timeline:

Submission Date:	25th Sep 24
Editorial Decision:	31st Oct 24
Revision Received:	2nd Feb 25
Editorial Decision:	24th Mar 25
Revision Received:	30th Mar 25
Accepted:	26th Apr 25

Editor: Poonam Bheda

Transaction Report:

31st Oct 2024

Manuscript Number: MSB-2024-12650

Title: LaGrACE: Estimating gene program dysregulation with latent regulatory networks

Dear Dr. Benos,

Thank you again for submitting your work to Molecular Systems Biology. We have now heard back from the three reviewers who agreed to evaluate your study. As you will see below, the reviewers appreciate that the proposed approach addresses a timely topic. However, they raise a series of concerns, which we would ask you to address in a major revision.

Without repeating all the comments listed below, some of the more fundamental issues raised are the following:

- benchmarking against other similar methods should be improved
- overstatements on improved performance should be tempered
- additional validation of the methods used in various contexts should be added

All other issues raised would need to be satisfactorily addressed. Please let me know in case you would like to discuss in further detail any of the comments, I would be happy to schedule a call.

We require:

1) A .docx formatted version of the manuscript text (including legends for main figures, EV figures and tables). Please make sure that the changes are highlighted to be clearly visible. Alternatively you may choose to submit your manuscript as a LaTeX file.

4) A .docx formatted letter INCLUDING the reviewers' reports and your detailed point-by-point responses to their comments. As part of the EMBO Press transparent editorial process, the point-by-point response is part of the Peer Review File (PRF), which will be published alongside your paper.

5) A complete author checklist, which you can download from our author guidelines (<https://www.embopress.org/page/journal/17574684/authorguide#submissionofrevisions>). Please insert information in the checklist that is also reflected in the manuscript. The completed author checklist will also be part of the PRF.

6) Please note that all corresponding authors are required to supply an ORCID ID for their name upon submission of a revised manuscript.

7) It is mandatory to include a 'Data Availability' section after the Materials and Methods. Before submitting your revision, primary datasets produced in this study need to be deposited in an appropriate public database, and the accession numbers and database listed under 'Data Availability'. Please remember to provide a reviewer password if the datasets are not yet public (see <https://www.embopress.org/page/journal/17574684/authorguide#dataavailability>).

This study includes no data deposited in external repositories.

8) All Materials and Methods need to be described in the main text using our 'Structured Methods' format, which is required for all research articles. According to this format, the Methods section includes a Reagents and Tools Table (listing key reagents, experimental models, software and relevant equipment and including their sources and relevant identifiers) followed by a Methods and Protocols section describing the methods using a step-by-step protocol format. The aim is to facilitate adoption of the methodologies across labs. Please upload the Reagents and Tools table as a separate document when submitting your revised manuscript. More information on how to adhere to this format as well as a downloadable template (.docx) for the Reagents and Tools Table can be found in our author guidelines:

<https://www.embopress.org/page/journal/17444292/authorguide#structuredmethods>

An example of a Method paper with Structured Methods can be found here:
<https://www.embopress.org/doi/10.15252/msb.20178071>.

9) For data quantification: please specify the name of the statistical test used to generate error bars and P values, the number (n) of independent experiments (specify technical or biological replicates) underlying each data point and the test used to calculate p-values in each figure legend. The figure legends should contain a basic description of n, P and the test applied. Graphs must include a description of the bars and the error bars (s.d., s.e.m.). Please provide exact p values.

10) Our journal encourages inclusion of *data citations in the reference list* to directly cite datasets that were re-used and obtained from public databases. Data citations in the article text are distinct from normal bibliographical citations and should directly link to the database records from which the data can be accessed. In the main text, data citations are formatted as follows: "Data ref: Smith et al, 2001" or "Data ref: NCBI Sequence Read Archive PRJNA342805, 2017". In the Reference list, data citations must be labeled with "[DATASET]". A data reference must provide the database name, accession number/identifiers and a resolvable link to the landing page from which the data can be accessed at the end of the reference. Further instructions are available at .

11) We replaced Supplementary Information with Expanded View (EV) Figures and Tables that are collapsible/expandable online. A maximum of 5 EV Figures can be typeset. EV Figures should be cited as 'Figure EV1, Figure EV2' etc... in the text and their respective legends should be included in the main text after the legends of regular figures.

<https://www.embopress.org/page/journal/17574684/authorguide#expandedview>

13) Author contributions: CRedit has replaced the traditional author contributions section because it offers a systematic machine readable author contributions format that allows for more effective research assessment. Please remove the Authors Contributions from the manuscript and use the free text boxes beneath each contributing author's name in our system to add specific details on the author's contribution. More information is available in our guide to authors.

Please also suggest a striking image or visual abstract to illustrate your article as a PNG file 550 px wide x 300-600 px high. Share synopsis text and image, as well as eTOC:

Please note that these would be the final versions and changes during proofing are usually not allowed

16) As part of the EMBO Publications transparent editorial process initiative (see our policy here:

https://www.embopress.org/transparent-process#Review_Process), Molecular Systems Biology will publish online a Peer Review File (PRF) to accompany accepted manuscripts.

In the event of acceptance, this file will be published in conjunction with your paper and will include the anonymous referee reports, your point-by-point response and all pertinent correspondence relating to the manuscript. Let us know whether you agree with the publication of the PRF and as here, if you want to remove or not any figures from it prior to publication.

Please note that the Authors checklist will be published at the end of the PRF.

Molecular Systems Biology has a "scooping protection" policy, whereby similar findings that are published by others during review or revision are not a criterion for rejection. Should you decide to submit a revised version, I do ask that you get in touch after three months if you have not completed it, to update us on the status.

I look forward to receiving your revised manuscript.

Yours sincerely,

Poonam Bheda, PhD
Scientific Editor
Molecular Systems Biology

Reviewer #1:

The problem addressed in this work is to classify/cluster transcriptomics samples based on the degree of dysregulation of transcriptional programs. Stated as such, this is a problem for which numerous tools have been previously published, based on a variety of ways to extract information from the transcriptomic profiles. Authors present a yet another way, which relies on (1) estimating latent factor and building a causal network among them from a 'reference' cohort of samples, (2) In a 'case' sample, estimating the baseline activity of a node/program/latent factor from the activity of the 'markov blanket' nodes (neighboring nodes in the causal network inferred from reference cohort), and then (3) estimating the Observed-Expected divergence of the activity. Given the vector of divergence values for all nodes, they apply standard tools of clustering to obtain the subtypes of the case samples. The main novel aspects of this approach are the estimation of causal network from reference and using it to estimate the expected baseline activity of a program in a case sample. From methodological perspective this is nice and makes sense. They demonstrate the value of their approach through applications in breast cancer, COPD and scRNA-seq drug perturbation data. However, the performance gain is sometime overstated, especially in COPD application the improvement is too small to be of any consequence, and the comparison with related methods is insufficient

Major

The biological rationale for focusing on gene expression programs instead of individual genes is well-founded. However, a more detailed explanation of the novelty and advantages of the proposed Bayesian approach compared to related gene expression program discovery methods e.g., scITD (PMID: 39313646), DIALOGUE (PMID: 35513526) is needed.

The authors demonstrate the utility of their approach for refining molecular subtypes of tumors. However, there are existing methods for tumor subtyping based on context-specific gene regulatory networks. A comparison of their approach to these prior methods (PMID: 21489222, PMID: 34880275, 30598111) is needed to establish novelty and relative advantage.

In the synthetic data validation, provide a result showing the effect of how distinct the perturbed networks are, in terms of number of altered edges.

Fig 2D,E suggest that almost all of the improvement could come from simply using the latent factor (as opposed the observed features) and a minor further improvement comes from Lagrace specifically. Authors do not sufficiently acknowledge this observation, thus overstating the relative contribution of Lagrace. In Fig 2C, please add the clustering by LFs. One presumes it would look as clean as Lagrace clusters.

"Cluster 0 notably grouped 82% of HER2+ and 29% of luminal B samples". Please compare the clinical features of HER2+ in cluster 0 and HER2+ in other clusters. Likewise for lumB. This will clarify the reasons for the split of HER2+ and LumB in multiple clusters.

Re. "Biological Networks Driving Breast Cancer Subtypes Networks - second paragraph". Focusing on luminal samples, specifically in cluster 1, please show that the cluster-specific factors are associated with survival while cluster-2,3-specific factors are not. Do the same for cluster-2 and cluster-3 samples to establish specificity.

"We validated this finding by inhibiting FOXM1 in breast cancer cell lines, and we observed reduced cell motility". It is not clear how this validation relates to a specific finding by LaGrace. Are they claiming that "FOXM1 link to tumor aggressiveness (proxied by motility)" is novel finding by Lagrace? Which is not the case.

"Cluster 0 and 3 represented the most impaired groups, showing poorer spirometry results (FEV1/FVC ratio)". This doesn't see true for cluster 0 in Fig 5C.

"more symptom (measured by COPD Assessment Test and SGRQ St. George's Respiratory Questionnaire) and dyspnea

(measured by Medical Research Council Dyspnea Scale)". This claim is not supported.

"displayed distinctive patterns in cluster 3,". This claim is not supported.

The difference in survival shown in Fig 5B seem too minor to be of any real consequence.

It is not clear how LEF1 transcriptome is obtained and how LEF1 motif is included on the network.

"Only the full-length LEF1 exhibited a positively correlation with FEV1 %predicted and FEV1/FVC ratio. This association was not observed for the partial region lacking the β -catenin-binding domain.". This data is not shown.

In COPD analysis, to control for age, they include individuals 55 years or older. Given that a majority of patients will qualify this, this seems like a very large range and doesn't really control for Age. A direct regression-based control ought to be done.

While the authors have benchmarked their approach using cancer cell line data, it would be beneficial to assess its effectiveness in more challenging clinical contexts. For example, disentangling factors linked to T cell exhaustion from those linked to T cell activation/stemness in patients receiving immunotherapy could be explored using single-cell RNA-Seq datasets (e.g., GSE176021). Comparing their results to prior work in this area (<https://www.nature.com/articles/s41587-023-01940-3#Sec2>) would provide further insights.

Minor

"Gene programs are typically estimated using factor analysis or NMF methods. However, these methods often face issues with identifiability and impose orthogonality constraints. NMF doesn't.". This is not true for NMF.

Better to order all main/suppl figures tables in the order they are referred to in the main text.

Please clarify the definition and calculation of purity scores used for benchmarking.

Ensure that the mathematical notations clearly imply a single divergence score for each gene expression program, rather than for the entire sample.

Reviewer #2:

This is a very-well written paper with detailed analysis. I have a few suggestions:

1. I would suggest adding a section on computational performance, including run-time analysis and scalability for large datasets compared to small datasets. This could help readers assess the feasibility of using LaGrACE for their own datasets.
2. The paper shows the technical success of LaGrACE in identifying clusters and stratifying patients. It would be great to expand the discussion on possible applications on precision medicine and how the findings would impact treatment outcomes.

Reviewer #3:

In this manuscript, the authors propose a novel approach LaGrACE, which estimates regulatory network-based pseudo control outcome to characterize gene program dysregulation. Given dysregulation of regulatory network can lead to a broad range of diseases, LaGrACE facilitates grouping of samples exhibiting similar patterns of gene program dysregulation. However, before this work is deemed suitable for publication, several points require further clarification:

1. How does the author ensure that the best latent factors are learned? Are these latent factors interpretable? I mean do they correspond to some biological entities? Additionally, how is the value of k determined when calculating the factor matrix through matrix factorization?
2. In complex biological systems, there are numerous feedback mechanisms among biological molecules and processes (biological processes is one type of latent factors in this manuscript). Given this, is it reasonable to model these relationships

using Directed Acyclic Graphs (DAGs)? Alternatively, in the presence of loops, is the construction of the Markov skeleton based on conditional independence still correct? It is suggested that the author conducts some derivations and proofs.

3. In the experiment of "Identifying Driving Networks of Breast Cancer Subtypes", is there a way to prove (both wet lab and dry lab experiments are acceptable) that this network indeed plays a decisive role in determining cancer subtypes?

POINT-BY-POINT RESPONSE TO REVIEWERS

Manuscript Number: MSB-2024-12650

Title: LaGrACE: Estimating gene program dysregulation with latent regulatory networks

Reviewer #1:

The problem addressed in this work is to classify/cluster transcriptomics samples based on the degree of dysregulation of transcriptional programs. Stated as such, this is a problem for which numerous tools have been previously published, based on a variety of ways to extract information from the transcriptomic profiles. Authors present a yet another way, which relies on (1) estimating latent factor and building a causal network among them from a 'reference' cohort of samples, (2) In a 'case' sample, estimating the baseline activity of a node/program/latent factor from the activity of the 'markov blanket' nodes (neighboring nodes in the causal network inferred from reference cohort), and then (3) estimating the Observed-Expected divergence of the activity. Given the vector of divergence values for all nodes, they apply standard tools of clustering to obtain the subtypes of the case samples. The main novel aspects of this approach are the estimation of causal network from reference and using it to estimate the expected baseline activity of a program in a case sample. From methodological perspective this is nice and makes sense. They demonstrate the value of their approach through applications in breast cancer, COPD and scRNA-seq drug perturbation data. However, the performance gain is sometime overstated, especially in COPD application the improvement is too small to be of any consequence, and the comparison with related methods is insufficient

Major

The biological rationale for focusing on gene expression programs instead of individual genes is well-founded. However, a more detailed explanation of the novelty and advantages of the proposed Bayesian approach compared to related gene expression program discovery methods e.g., scITD (PMID: 39313646), DIALOGUE (PMID: 35513526) is needed.

Response:

We thank the reviewer for this comment. We have now revised the Introduction to highlight the differences between LaGrACE and these methods. Briefly:

- DIALOGUE [1] combines penalized matrix decomposition with multilevel modeling to identify multicellular programs (MCPs) of co-regulated genes across different cell types. Each MCP consists of multiple cell-type-specific gene programs, and these MCPs are designed to be **mutually uncorrelated**.
- Similarly, scITD [2] employs Tucker tensor decomposition to infer multicellular gene expression programs that vary across donors or samples. scITD utilizes higher-order

orthogonal iteration (HOOI) for tensor decomposition, ensuring that the inferred gene programs are distinct and **orthogonal**.

In contrast, our method, LaGrACE, introduces a novel approach to estimating gene program dysregulation by first inferring latent factors representing gene programs and subsequently constructing a regulatory network to capture interactions among these gene programs. Unlike DIALOGUE and scITD, LaGrACE requires non-orthogonal gene programs to construct the reference network and allows overlap in the observed features between latent factors. To achieve this, it leverages **LOVE** [3], which offers two key advantages: **(1)** theoretical guarantees for the unique identifiability of latent factors without relying on assumptions about the data-generating mechanisms, and **(2)** the ability to identify non-orthogonal latent factors without imposing restrictive orthogonality constraints.

References for this comment:

1. Jerby-Aron, Livnat, and Aviv Regev. "DIALOGUE maps multicellular programs in tissue from single-cell or spatial transcriptomics data." *Nature biotechnology* vol. 40,10 (2022): 1467-1477. doi:10.1038/s41587-022-01288-0
2. Mitchel, Jonathan et al. "Coordinated, multicellular patterns of transcriptional variation that stratify patient cohorts are revealed by tensor decomposition." *Nature biotechnology*, 10.1038/s41587-024-02411-z. 23 Sep. 2024, doi:10.1038/s41587-024-02411-z
3. Bing, Xin et al. "Latent Model-Based Clustering for Biological Discovery." *iScience* vol. 14 (2019): 125-135. doi:10.1016/j.isci.2019.03.018

The authors demonstrate the utility of their approach for refining molecular subtypes of tumors. However, there are existing methods for tumor subtyping based on context-specific gene regulatory networks. A comparison of their approach to these prior methods (PMID: 21489222, PMID: 34880275, 30598111) is needed to establish novelty and relative advantage.

Response:

This is a very interesting suggestion. In general, comparison of subtyping methods is difficult when ground truth is not available. However, we looked into these studies hoping to be able to compare them at least in terms of survival discrimination. Unfortunately, we were not able to perform these comparisons due to the unavailability of both the code and data/metadata, like patient-level cluster information, that were used in the original studies. Detailed explanation follows:

Study [1] (PMID: 21489222): This study presents an unsupervised computational approach for dissecting breast cancer heterogeneity by identifying context-specific gene regulatory networks using the ExPattern algorithm. It would have been great to compare LaGrACE with this method as it seems to be the closest to ours. Unfortunately, the algorithm is no longer accessible as the provided download link (<http://sysbio.fulton.asu.edu/expattern>) is inactive. [Code and patient-level metadata are not available]

Study [2] (PMID: 34880275): This method focuses on cancer subtyping by constructing personalized gene regulatory networks for individual patients. It employs Bayesian networks to estimate gene regulatory networks, calculates edge contribution values (ECv) for each patient, and then clusters patients based on the ECv matrix. constructs a Bayesian network with B-spline nonparametric regression directly on gene expression data. However, the high dimensionality and multicollinearity of gene expression data may significantly impact Bayesian network estimation and introduce bias in subsequent steps [3,4]. Still, a direct comparison in real data would have been useful. Unfortunately, the code necessary to reproduce their results, particularly for ECv calculation [5], is not available. [Code and patient-level metadata are not available]

Study [6] (PMID: 30598111): This study proposes a cancer subtype prediction method that integrates multi-source expression data and heterogeneous biological regulatory networks, using mRNA expression, TF expression, miRNA expression, and the associations between them (mRNA-TF-miRNA regulatory networks) taken from prior knowledge. By contrast, LaGrACE follows an agnostic procedure. The gene programs (i.e., latent factors) it extracts automatically are general. Some latent factors may correspond to gene regulation by TF activity (more accurate than depending on TF mRNA expression due to poor concurrence sometimes), miRNA activity, chromatin accessibility or any other unmeasured factor. Again, it would have been interesting to test whether these theoretical advantages of LaGrACE translate into superior performance in practice. Unfortunately, the processed data [7] and (prior) regulatory network datasets [7] used in this study are not available, as the download link [8] is inactive. [Prior knowledge needed to run this method is not available; patient-level metadata are not available]

While these studies offer valuable insights into cancer subtyping, the lack of access to their underlying data or code unavailability limit our ability to perform a direct comparison. Regardless, we believe our approach provides a distinct and innovative contribution by refining molecular subtypes of tumors using a novel methodology, which we have thoroughly detailed in our work.

References for this comment:

1. Nasser, Sara et al. "Context-specific gene regulatory networks subdivide intrinsic subtypes of breast cancer." *BMC bioinformatics* vol. 12 Suppl 2,Suppl 2 S3. 29 Mar. 2011, doi:10.1186/1471-2105-12-S2-S3
2. Nakazawa, Mai Adachi et al. "Novel cancer subtyping method based on patient-specific gene regulatory network." *Scientific reports* vol. 11,1 23653. 8 Dec. 2021, doi:10.1038/s41598-021-02394-w
3. Al-Essa, Laila A., Endris Assen Ebrahim, and Yusuf Ali Mergiaiw. "Bayesian regression modeling and inference of energy efficiency data: the effect of collinearity and sensitivity analysis." *Frontiers in Energy Research*, 12 (2024): 1416126.
4. Kalnins, Arturs. "Multicollinearity: How common factors cause Type 1 errors in multivariate regression." *Strategic Management Journal* 39.8 (2018): 2362-2385.].
5. Tanaka, Yoshihisa et al. "System-Based Differential Gene Network Analysis for Characterizing a Sample-Specific Subnetwork." *Biomolecules* vol. 10,2 306. 14 Feb. 2020, doi:10.3390/biom10020306

6. Guo, Yang et al. "Improvement of cancer subtype prediction by incorporating transcriptome expression data and heterogeneous biological networks." BMC medical genomics vol. 11,Suppl 6 119. 31 Dec. 2018, doi:10.1186/s12920-018-0435-x
7. Xu, Taosheng et al. "Identifying Cancer Subtypes from miRNA-TF-mRNA Regulatory Networks and Expression Data." PloS one vol. 11,4 e0152792. 1 Apr. 2016, doi:10.1371/journal.pone.0152792
8. URL: <http://nugget.unisa.edu.au/Thuc/cancersubtypes/>

In the synthetic data validation, provide a result showing the effect of how distinct the perturbed networks are, in terms of number of altered edges.

Response:

We simulated a reference network with an average vertex degree of 2 and created three perturbed networks by introducing random edge modifications (15% of the total edges, including both deletions and additions). Each network consisted of 50 continuous variables and 25 discrete variables (75 edges per graph), where the discrete variables had four categorical levels. To ensure a robust evaluation, we generated 10 datasets for analysis. The table below summarizes the number of altered edges between the reference network and each of the perturbed networks across the datasets:

ref vs graph 1	ref vs graph 2	ref vs graph 3	graph 1 vs graph 2	graph 1 vs graph 3	graph 2 vs graph 3
11	11	11	20	20	21
11	11	11	19	19	20
10	11	11	21	18	22
11	11	11	19	22	21
11	11	11	21	21	21
11	11	11	19	22	21
11	11	11	19	20	20
11	11	11	21	19	22
11	11	11	22	21	22
11	11	11	21	22	21

Fig 2D,E suggest that almost all of the improvement could come from simply using the latent factor (as opposed the observed features) and a minor further improvement comes from Lagrace specifically. Authors do not sufficiently acknowledge this observation, thus overstating the relative contribution of Lagrace. In Fig 2C, please add the clustering by LFs. One presumes it would look as clean as Lagrace clusters.

Response:

We apologize for the confusion, which was caused by a mislabeling error. Before we provide a detailed response on how we corrected this, please let us clarify few things about Fig. 2.

1. In Fig. 2B, we show how LaGrACE strategy (without latent factor extraction) improves clustering compared to clustering using directly the observed features. (in low-dimensional space). We call those results “LaGrACE Continuous/Mixed” and “Observed Continuous”, respectively. This shows that LaGrACE methodology offers a significant advantage over clustering using the observed data directly.
2. In Fig. 2E (previously: Fig. 2D) we compare the clustering performance of LaGrACE in synthetic data using the known latent factors (LFs), which was used to create the observed data and the full LaGrACE method, where LFs are *inferred* using LOVE. We refer to the known LFs as “Ground Truth LF”.

Now, regarding the comment of the reviewer, in the first version of the paper, we mislabeled the x-axis of the violin plot in Figure 2D (now Fig. 2E) as 'Continuous Variable,' 'Ground Truth LF,' 'LaGrACE LOVE,' and 'LaGrACE Ground Truth LF'. The correct labels should have been:

- 'Continuous Variable'
- 'LaGrACE Ground Truth LF Continuous'
- 'LaGrACE LOVE Mixed'
- 'LaGrACE Ground Truth LF Mixed'

These are in accordance with the explanation we gave above. We have now relabeled the figure and provided further explanation in the caption of Fig. 2. The clustering using only LFs is also provided in the UMAP Figure below.

Regarding clustering performance, the results for '**LaGrACE Ground Truth LF Continuous**' in (now) Figure 2E align with the clustering performance of '**LaGrACE Continuous**' shown in Figure 2B. This is expected because the high-dimensional simulation data used in (now) Figure 2E was generated based on the same low-dimensional data used for Figures 2A and 2B. Minor discrepancies in clustering results arise from the use of different random seed values for clustering. To maintain consistency, we will refer to the results from '**LaGrACE Ground Truth LF Continuous**' as shown in Figure 2B.

To provide further information to the reviewer regarding this comment, we have included below updated visualizations that summarize clustering performance (ARI and AMI) across five conditions:

1. High-dimensional (2500) continuous variables.
2. Ground truth latent factors (50 continuous variables).
3. LaGrACE features inferred from ground truth latent factors.
4. LaGrACE features inferred using LOVE latent factors and discrete variables.
5. LaGrACE features inferred from ground truth latent factors combined with discrete variables.

These updates confirm that LaGrACE significantly enhances clustering performance, with the inclusion of discrete variables contributing a minor but consistent additional improvement.

"Cluster 0 notably grouped 82% of HER2+ and 29% of luminal B samples". Please compare the clinical features of HER2+ in cluster 0 and HER2+ in other clusters. Likewise for lumB. This will clarify the reasons for the split of HER2+ and LumB in multiple clusters.

Response:

We thank the reviewer for the insightful suggestion. We analyzed the clinical features of **Luminal B** and **HER2+** samples across LaGrACE clusters to better understand the reasons for their separation into multiple clusters.

For **Luminal B** samples, the LaGrACE-defined clusters are significantly associated with the following clinical features:

- **Disease-specific survival** (p-value = 0.0384)
- **Distal relapse** (p-value = 0.0142)
- **Nottingham Prognostic Index (NPI)** (p-value = 0.00014)

These findings indicate that the stratification of Luminal B samples into distinct clusters may capture clinically relevant heterogeneity, particularly with respect to disease prognosis and progression. In other words, it seems that LaGrACE effectively identifies subgroup-specific variations within the Luminal B subtype.

For **HER2+** samples, the LaGrACE clusters are significantly associated with the **Nottingham Prognostic Index (NPI)**(p-value = 0.017). These results also emphasize the utility of LaGrACE in uncovering clinically meaningful heterogeneity within breast cancer subtypes.

We have now added these results as Appendix Fig. S5.

Re. "Biological Networks Driving Breast Cancer Subtypes Networks - second paragraph". Focusing on luminal samples, specifically in cluster 1, please show that the cluster-specific factors are associated with survival while cluster-2,3-specific factors are not. Do the same for cluster-2 and cluster-3 samples to establish specificity.

Response:

We thank the reviewer for the suggestion. We agree that survival-specific factors provide a more nuanced understanding of disease-associated molecular mechanisms than tumor size alone. Following the reviewer's suggestion, we focused on identifying factors specifically contributing to survival within each cluster, using Cox regression. We briefly describe the results below, which we have now added as **Appendix Table S4**.

We identified several factors enriched for biological pathways that were specifically associated with survival in each cluster:

Galactose Metabolism (Cluster 1)

- Cluster 1: HR = 0.58, p-value = **4.7E-04**
- Cluster 2: HR = 0.76, p-value = 0.14
- Cluster 3: HR = 0.99, p-value = 0.97

Galactose metabolism was significantly associated with survival in Cluster 1 only. This pathway has been linked to breast cancer survival and metastasis [1, 2].

Integrin-Linked Kinase (ILK) Signaling (Cluster 2)

- Cluster 1: HR = 1.09, p-value = 0.66
- Cluster 2: HR = 1.27, p-value = **7.77E-04**
- Cluster 3: HR = 1.16, p-value = 0.12

ILK signaling was specifically associated with survival in Cluster 2 only. This pathway is linked to breast cancer therapeutic vulnerability and metastatic progression. [3, 4, 5].

E2F Targets-2 (Cluster 3)

- Cluster 1: HR = 1.13, p-value = 0.50
- Cluster 2: HR = 1.16, p-value = 0.20
- Cluster 3: HR = 0.77, p-value = **0.022**

This factor was associated with survival specifically in Cluster 3 only. It also showed an association with tumor size for Cluster 3 samples. E2F is a well-known regulator of cell cycle progression and proliferation, contributing to cancer progression.

Regulation of RhoA Activity (Cluster 3)

- Cluster 1: HR = 1.07, p-value = 0.67
- Cluster 2: HR = 1.20, p-value = 0.10
- Cluster 3: HR = 1.68, p-value = **4.44E-05**

Regulation of RhoA activity was significantly associated with survival in Cluster 3 only. RhoA, a Rho GTPase, is implicated in breast cancer metastasis [6, 7].

These cluster-specificities underscore LaGrACE's ability to elucidate disease-associated molecular mechanisms tailored to each subgroup, providing valuable insights into disease heterogeneity.

References for this comment:

1. Young, C Megan et al. "Metabolic dependencies of metastasis-initiating cells in female breast cancer." *Nature communications* vol. 14,1 7076. 4 Nov. 2023, doi:10.1038/s41467-023-42748-8
2. Han, Xiangchen et al. "Integrated Multi-Omics Profiling of Young Breast Cancer Patients Reveals a Correlation between Galactose Metabolism Pathway and Poor Disease-Free Survival." *Cancers* vol. 15,18 4637. 19 Sep. 2023, doi:10.3390/cancers15184637
3. McDonald, Paul C, and Shoukat Dedhar. "New Perspectives on the Role of Integrin-Linked Kinase (ILK) Signaling in Cancer Metastasis." *Cancers* vol. 14,13 3209. 30 Jun. 2022, doi:10.3390/cancers14133209
4. Hinton, Cimina V et al. "Contributions of integrin-linked kinase to breast cancer metastasis and tumorigenesis." *Journal of cellular and molecular medicine* vol. 12,5A (2008): 1517-26. doi:10.1111/j.1582-4934.2008.00300.x
5. Beetham, Henry et al. "Loss of Integrin-Linked Kinase Sensitizes Breast Cancer to SRC Inhibitors." *Cancer research* vol. 82,4 (2022): 632-647. doi:10.1158/0008-5472.CAN-21-0373
6. Humphries, Brock et al. "Rho GTPases: Big Players in Breast Cancer Initiation, Metastasis and Therapeutic Responses." *Cells* vol. 9,10 2167. 25 Sep. 2020, doi:10.3390/cells9102167
7. Mohammadipour, Amina et al. "RhoA-ROCK competes with YAP to regulate amoeboid breast cancer cell migration in response to lymphatic-like flow." *FASEB bioAdvances* vol. 4,5 342-361. 14 Feb. 2022, doi:10.1096/fba.2021-00055

"We validated this finding by inhibiting FOXM1 in breast cancer cell lines, and we observed reduced cell motility". It is not clear how this validation relates to a specific finding by LaGrace. Are they claiming that "FOXM1 link to tumor aggressiveness (proxied by motility)" is novel finding by Lagrace? Which is not the case.

Response:

We apologize for the confusion. Our LaGrACE-defined clusters identified three distinct luminal subgroups (clusters 1, 2, and 3). Within these clusters, we observed that E2F target genes, including FOXM1, were downregulated in Cluster 1, which exhibits a lower metastasis rate and higher overall survival compared to the other luminal clusters. This association suggested that reduced FOXM1 activity might be linked to less aggressive tumor behavior.

To investigate whether downregulating FOXM1 could influence metastatic potential, we conducted an experimental validation using breast cancer cell lines derived from different sources, such as pleural effusions (MDA-MB-231) and primary breast tumors (SUM149 and SUM159). We treated these cells with three different small-molecule FOXM1 inhibitors—Thiostrepton, FDI-6, and RCM-1—each targeting FOXM1 through distinct mechanisms. All three inhibitors effectively reduced cell motility in these breast cancer cell lines.

We do not claim that FOXM1's link to tumor aggressiveness is novel in itself; FOXM1's role in cancer progression is well-documented. Instead, our findings illustrate that LaGrACE can help prioritize and highlight informative biomarkers and pathways within specific tumor subtypes. By identifying FOXM1 as a potentially actionable factor in the less aggressive cluster, LaGrACE demonstrates its value as a tool for pinpointing targets that may guide therapeutic strategies or serve as biomarkers.

We have now added text to "Identification of Biological Networks Driving Breast Cancer Subtypes Networks" to clarify this.

"Cluster 0 and 3 represented the most impaired groups, showing poorer spirometry results (FEV1/FVC ratio)". This doesn't see true for cluster 0 in Fig 5C.

"more symptom (measured by COPD Assessment Test and SGRQ St. George's Respiratory Questionnaire) and dyspnea (measured by Medical Research Council Dyspnea Scale)". This claim is not supported.

Response:

We thank the reviewer for catching this. The reviewer is right that not all those measures are similarly worse for *both* clusters 0 and 3. We have now modified the text to clearly state which of these features show worse outcomes for cluster 0 and which for cluster 3.

But the overall assessment is right: both clusters show poorer spirometry (but measured differently: FEV1/FVC, FEV1 %predicted for cluster 0; FVC %predicted for cluster 3), both show worse 6-min walk results and higher SGRQ symptom burden and dyspnea scores (data presented in **Figure 5B, Appendix Figure S11B, Table 2**).

We also note in the revised version that cluster 0 consists of predominantly non-Hispanic white (NHW) participants, while cluster 3 consists predominantly of African American (AA) participants. While the FEV1/FVC ratio is a common measure to diagnose COPD and estimate its severity, it can be influenced by race, ethnicity, age, sex, and body size. Indeed, the use of a fixed FEV1/FVC ratio as a diagnostic criterion can underdiagnose COPD in African American participants [1]. Furthermore, race-specific spirometry references have not improved prediction of breathlessness or prognosis [2, 3].

Below we present some of the relevant statistics from Table 2 (selected symptom variables), which in the paper contains all results.

Table 2 (selected variables): **Clinical characteristics of COPD participants vary across clusters.** P-values were calculated using a Kruskal-Wallis test for continuous and ordinal variables, and a Chi-squared test for discrete variables, to assess if there are differences in the distribution of variables among clusters.

Variable	Reference Group	Cluster 0	Cluster 1	Cluster 2	Cluster 3	# missing	P.Val
Participants, n	139	556	478	357	220	0	
Age at Phase 2 visit, yr, mean (SD)	64.527 (7.544)	69.182 (8.034)	68.867 (8.375)	69.832 (8.045)	62.075 (7.413)	0	
gender, Female, n (%)	78 (56.12)	224 (40.29)	230 (48.12)	136 (38.1)	110 (50)	0	
race, African American, n (%)	29 (20.86)	60 (10.79)	31 (6.49)	28 (7.84)	217 (98.64)	0	
race, White, n (%)	110 (79.14)	527 (94.78)	447 (93.51)	329 (92.16)	3 (1.36)	0	
Symptoms							
MMRC dyspnea score, mean (SD)	0.626 (1.031)	1.766 (1.459)	1.454 (1.389)	1.529 (1.468)	1.95 (1.469)	0	2.66E-05
CAT score, mean (SD)	7.36 (5.992)	14.959 (8.842)	13.134 (8.526)	13.958 (8.351)	16.086 (9.092)	0	7.58E-05
Six-minute walk distance, ft, mean (SD)	1530.109 (408.52)	1174.192 (465.711)	1258.888 (425.263)	1211.714 (439.033)	1132.23 (438.257)	39	1.38E-03
SGRQ score: Total, mean (SD)	12.548 (13.811)	32.415 (21.92)	28.492 (21.018)	30.077 (21.368)	34.1 (22.435)	0	4.14E-03

CAT COPD Assessment Test, MMRC Modified Medical Research Council Dyspnea Scale, SGRQ St. George's Respiratory Questionnaire

References for this comment:

1. Regan, Elizabeth A et al. "Use of the Spirometric "Fixed-Ratio" Underdiagnoses COPD in African-Americans in a Longitudinal Cohort Study." *Journal of general internal medicine* vol. 38,13 (2023): 2988-2997. doi:10.1007/s11606-023-08185-5
2. Ekström, Magnus, and David Mannino. "Research race-specific reference values and lung function impairment, breathlessness and prognosis: Analysis of NHANES 2007-2012." *Respiratory research* vol. 23,1 271. 1 Oct. 2022, doi:10.1186/s12931-022-02194-4
3. Regan, Elizabeth A et al. "Early Evidence of Chronic Obstructive Pulmonary Disease Obscured by Race-Specific Prediction Equations." *American journal of respiratory and critical care medicine* vol. 209,1 (2024): 59-69. doi:10.1164/rccm.202303-0444OC

"displayed distinctive patterns in cluster 3,". This claim is not supported.

Response:

We might not have explained this statement adequately, but this sentence was meant to highlight the distinctive patterns in immune cell distributions observed in Cluster 3. The support of this claim can be found in **Table 2** and **Appendix Figure S12**, where we report the complete blood count measurements, including neutrophil and lymphocyte percentages. Compared with the other clusters—primarily composed of non-Hispanic White participants—Cluster 3 showed a higher lymphocyte percentage and a lower neutrophil percentage, resulting in the lowest neutrophil-to-lymphocyte ratio among the four clusters.

In response to reviewers' comment, we have now added the reference to **Table 2** and **Appendix Fig. S12** in the main text.

Also, to clarify this further, we have now included the neutrophil-to-lymphocyte ratio in **Table 2**. The selected complete blood count variables highlighted below (and detailed in the revised manuscript) support our claim that Cluster 3 exhibits distinctive immune-related patterns.

Table 2 (selected variables): **Clinical characteristics of COPD participants vary across clusters.** P-values were calculated using a Kruskal-Wallis test for continuous and ordinal variables, and a Chi-squared test for discrete variables, to assess if there are differences in the distribution of variables among clusters.

Variable	Reference Group	Cluster 0	Cluster 1	Cluster 2	Cluster 3	# missing	P.Val
Participants, n	139	556	478	357	220	0	
Age at Phase 2 visit, yr, mean (SD)	64.527 (7.544)	69.182 (8.034)	68.867 (8.375)	69.832 (8.045)	62.075 (7.413)	0	
gender, Female, n (%)	78 (56.12)	224 (40.29)	230 (48.12)	136 (38.1)	110 (50)	0	
race, African American, n (%)	29 (20.86)	60 (10.79)	31 (6.49)	28 (7.84)	217 (98.64)	0	
race, White, n (%)	110 (79.14)	527 (94.78)	447 (93.51)	329 (92.16)	3 (1.36)	0	
Complete blood count measurement							
Lymphocyte percentage, mean (SD)	32.266 (9.522)	22.016 (7.14)	28.912 (8.277)	26.764 (8.072)	36.914 (9.046)	5	1.34E-89
Neutrophil percentage, mean (SD)	56.417 (9.616)	66.959 (8.596)	58.895 (8.66)	61.138 (8.856)	51.645 (10.072)	5	1.24E-85
Neutrophil to Lymphocyte ratio percentage, mean (SD)	2.034 (1.184)	3.665 (2.312)	2.345 (1.267)	2.679 (1.583)	1.622 (1.473)	7	1.30E-90
Neutrophil, K/uL, mean (SD)	3.792 (1.504)	5.216 (1.833)	4.404 (1.391)	4.625 (1.523)	3.327 (1.429)	7	8.31E-48
Lymphocytes, K/uL, mean (SD)	2.062 (0.661)	1.636 (0.578)	2.121 (0.774)	1.985 (0.775)	2.256 (0.718)	7	5.88E-38

The difference in survival shown in Fig 5B seem too minor to be of any real consequence.

Response:

We thank the reviewer for this comment, which made us re-examine our data. It is true that in the previous version of the figure, the survival curves seemed to converge towards the end, giving a borderline significant value of $p=0.016$. In that analysis, we had grouped COPDGene samples based on gene expression data collected during Phase 2 (2012–2017), and survival outcomes were assessed using data updated through August 2020 (Phase 3, 2018–2022). We note that this timeframe overlaps with the COVID-19 pandemic, a factor recorded in the COPDGene

timeline. The pandemic is likely to have affected mortality rates across all groups, causing the survival curves to converge and potentially obscuring differences attributable solely to COPD.

To address this concern, we re-examined survival data updated as of October 2018, prior to the pandemic. Using this earlier dataset, we observed more pronounced separation among the survival curves, revealing clearer differences in survival outcomes across the clusters. These results show cluster 0 to clearly have the worse prognosis ($p=0.009$; see figure below).

We have updated the manuscript to include these in order to provide a more accurate depiction of survival differences. The above plot is now included in **Fig. 5**.

It is not clear how LEF1 transcriptome is obtained and how LEF1 motif is included on the network.

Response:

It seems like this section needs more clarification. In our approach, LaGrACE infers a reference network that captures interactions among gene programs and potential clinical variables. This network is then used to compute pseudo-control outcomes and divergence scores. Figure 5G (previously Fig. 5D) depicts a subgraph of this reference network for the COPD samples, where each node represents a gene program. These gene programs are annotated using gene set enrichment analysis performed through Enrichr.

We identified two gene programs enriched for lymphocyte-related signatures: one associated with lymphocyte percentage (measured from complete blood count), and another enriched in CD4-specific genes, which correlated with CD4 T cell fractions (**Figure 5DE**). Within this subgraph, two additional gene programs are connected to these lymphocyte- and CD4-related nodes, and are enriched for binding of TCF/CTNNB1 (β -catenin) target gene promoters and LEF1 motif. LEF1 is

a transcription factor that, when binding to β -catenin, regulates lymphopoiesis via the canonical WNT pathway. It is known to promote CD4 lineage commitment through the induction of Th-POK, a lineage-specifying factor. LaGrACE was able to identify these high-level interactions, and the predicted network connectivity and enrichment analysis prompted us to investigate LEF1 transcripts and their association with COPD disease progression.

"Only the full-length LEF1 exhibited a positively correlation with FEV1 %predicted and FEV1/FVC ratio. This association was not observed for the partial region lacking the β -catenin-binding domain.". This data is not shown.

Response:

We might not have been clear enough here. Although we actually presented those results in **Table 3** (ECLIPSE cohort), we didn't properly reference this in the text. We now made sure that proper reference of Table 3 is included after this statement.

To further validate this association, we examined blood-derived gene expression data from the ECLIPSE study, which reported two distinct LEF1 isoforms: **LEF1 Full Length (AF288571)** and **LEF1 Partial Region (AF294627)**. Upon re-examination, we corrected a previous oversight: the AF294627 isoform lacks the high-mobility group (HMG) domain, not the β -catenin-binding domain [1]. Both the HMG domain and the β -catenin-binding domain are integral for WNT signaling mediated by TCF/LEF transcription factors [2–4]. The HMG domain is essential for DNA binding at Wnt-response elements, while the N-terminal domain interacts with β -catenin to activate Wnt target gene transcription.

The association analyses between the LEF1 isoforms and spirometry measurements (FEV1 %predicted and FEV1/FVC ratio) from the ECLIPSE dataset are presented in **Table 3** (included below). The results indicate that **only the full-length LEF1 isoform (AF288571)**, which retains the HMG domain, exhibits a positive correlation with pulmonary function measures. This finding underscores the functional importance of the HMG domain in LEF1-mediated WNT signaling and its relationship to lung function.

References for this comment:

1. Kobiela, A et al. "A novel isoform of human lymphoid enhancer-binding factor-1 (LEF-1) gene transcript encodes a protein devoid of HMG domain and nuclear localization signal." *Acta biochimica Polonica* vol. 48,1 (2001): 221-6.
2. Hoppler, Stefan, and Claire Louise Kavanagh. "Wnt signalling: variety at the core." *Journal of cell science* vol. 120, Pt 3 (2007): 385-93. doi:10.1242/jcs.03363
3. Söderholm, Simon, and Claudio Cantù. "The WNT/ β -catenin dependent transcription: A tissue-specific business." *WIREs mechanisms of disease* vol. 13,3 (2021): e1511. doi:10.1002/wsbm.1511

4. Cadigan, K.M. and Waterman, M.L. "TCF/LEFs and Wnt signaling in the nucleus." Cold Spring Harbor Perspectives in Biology vol. 4,11 a007906 (2012).

Table 3. Correlation Analysis between LEF1 gene expression and clinical variables. PCC represents Pearson correlation coefficient; SRCC represents Spearman's rank correlation coefficient.

Correlation Analysis	Variable	PCC (r)	P-val	SRCC (rho)	P-val
LaGrACE divergence score of LF: Lymphocyte percentage	FEV1/FVC (n=1749)	0.12	1.34E-06	0.10	6.05E-05
	FEV1 %Predicted (n=1750)	0.10	1.22E-04	0.08	1.18E-03
LaGrACE divergence score of LF: CD4-specific Genes	FEV1/FVC (n=1749)	0.20	8.68E-16	0.19	7.08E-14
	FEV1 %Predicted (n=1750)	0.11	1.41E-05	0.11	1.52E-05
COPDgene: LEF1 gene expression	Age (n=1750)	-0.31	7.81E-40	-0.32	3.20E-43
	FEV1/FVC (n=1749)	0.20	1.17E-16	0.19	3.19E-15
	FEV1 %Predicted (n=1750)	0.18	7.10E-14	0.16	9.53E-12
ECLIPSE: LEF1 partial region (AF294627)	Age (n=229)	-0.17	8.22E-03	-0.24	2.72E-04
	FEV1/FVC (n=229)	0.04	5.15E-01	0.09	1.96E-01
	FEV1 %Predicted (n=229)	0.04	5.58E-01	0.07	2.69E-01
ECLIPSE: LEF1 Full Length (AF288571)	Age (n=229)	-0.25	1.51E-04	-0.25	1.19E-04
	FEV1/FVC (n=229)	0.25	9.95E-05	0.27	4.66E-05
	FEV1 %Predicted (n=229)	0.24	1.94E-04	0.25	1.25E-04
Partial Correlation Analysis	Variable	PCC (r)	P-val	SRCC (rho)	P-val
COPDgene: LEF1 gene expression condition on AGE	FEV1/FVC (n=1749)	0.16	4.54E-11	0.15	1.11E-09
	FEV1 %Predicted (n=1750)	0.18	3.17E-14	0.16	1.55E-11
ECLIPSE: LEF1 partial region (AF294627) condition on AGE	FEV1/FVC (n=229)	0.00	9.70E-01	0.04	5.41E-01
	FEV1 %Predicted (n=229)	0.00	9.92E-01	0.04	5.85E-01
	FEV1/FVC (n=229)	0.20	2.11E-03	0.23	5.33E-04

ECLIPSE: LEF1 Full Length (AF288571) condition on AGE	FEV1 %Predicted (n=229)	0.20	2.38E-03	0.22	8.24E-04
--	-------------------------	-------------	-----------------	-------------	-----------------

In COPD analysis, to control for age, they include individuals 55 years or older. Given that a majority of patients will qualify this, this seems like a very large range and doesn't really control for Age. A direct regression-based control ought to be done.

Response:

There seems to be a confusion here. Please let us clarify. In our COPD analysis, we utilized 3 datasets:

1. COPDGene Study: Bulk RNA-sequencing data (blood-derived)
2. ECLIPSE Study: Bulk gene expression data (blood-derived)
3. Single-cell RNA-seq Data: Lung tissue samples from healthy controls and COPD patients

For the two bulk gene expression datasets (COPDGene and ECLIPSE), we accounted for the effects of aging by performing partial correlation analysis, as shown in **Table 3**, which is an alternative method to what the reviewer suggests. This approach helps isolate the influence of gene expression changes from the variance attributable to age.

For the single-cell RNA-seq dataset, we focused on how LEF1 regulon activity differs in T lymphocytes between these conditions. Current literature suggests that aging exerts heterogeneous and dynamic effects at the cellular level, rather than following a simple linear pattern [1–3]. Thus, applying a direct regression-based approach for age adjustment may not fully capture these complex, non-linear aging trajectories at the single-cell level.

Nonetheless, to address the reviewer's concern, we tested the robustness of our findings by removing the oldest individual from the control group. Originally, the control group (10 samples aged 55 years and older) had a mean age of 65.3 years (SD = 6.13), and the COPD group (17 samples aged 55 years and older) had a mean age of 61.8 years (SD = 4.27). After removing the oldest control subject, the control group's mean age shifted to 63.7 years (SD = 3.5), reducing the age difference between the two groups. Even after this adjustment, we still observed significantly downregulated LEF1 regulon scores in T lymphocytes from COPD patients (see next figure).

References for this comment:

1. Zhang, Zehao et al. “A panoramic view of cell population dynamics in mammalian aging.” *Science (New York, N.Y.)*, eadn3949. 28 Nov. 2024, doi:10.1126/science.adn3949
2. Shen, Xiaotao et al. “Nonlinear dynamics of multi-omics profiles during human aging.” *Nature aging* vol. 4,11 (2024): 1619-1634. doi:10.1038/s43587-024-00692-2
3. Jia, Minxue et al. “Transcriptional changes of the aging lung.” *Aging cell* vol. 22,10 (2023): e13969. doi:10.1111/accel.13969

While the authors have benchmarked their approach using cancer cell line data, it would be beneficial to assess its effectiveness in more challenging clinical contexts. For example, disentangling factors linked to T cell exhaustion from those linked to T cell activation/stemness in patients receiving immunotherapy could be explored using single-cell RNA-Seq datasets (e.g., GSE176021). Comparing their results to prior work in this area (<https://www.nature.com/articles/s41587-023-01940-3#Sec2>) would provide further insights.

We thank the reviewer for this insightful suggestion. Unfortunately, our algorithm is not designed for such cases, since these datasets lack a set of reference samples.

Our method primarily focuses on estimating gene program dysregulation based on a reference regulatory graph, which calculates from the data (no path information is needed). In this respect, it differs fundamentally from prior knowledge-based gene program approaches, such as Spectra, which directly utilize predefined gene sets (e.g., CD8 T cell tumor reactivity gene programs). Instead, LaGrACE employs an unsupervised data-driven approach to infer latent factors or gene programs, making the inferred factors potentially data-dependent and adaptable to the specific dataset under analysis.

With that in mind, and following the reviewer’s comment, we applied LaGrACE to the single-cell RNA-Seq dataset **GSE176021** [1], a lung cancer atlas of tumor-infiltrating T cells, in which TCR antigen specificity was functionally validated. T cells play a crucial role in antitumor immunity by

recognizing mutation-associated neoantigens (MANAs) and initiating immune responses leading to tumor cell apoptosis. We inferred latent factors and found one factor that specifically characterized **CD8+ T cells** with **MANA-specific TCRs** (visualized in the figure below). This demonstrates that LaGrACE can effectively uncover latent factors associated with tumor reactivity.

Among the 32 genes contributing to this latent factor, we identified key markers such as **ENTPD1**, **CXCL13**, and **HAVCR2**, which are known to play critical roles in T cell activation and exhaustion [2,3]. Notably, **ENTPD1** and **CXCL13** are part of the recently developed **MANAscore** gene program, which characterizes MANA-specific CD8+ tumor-infiltrating lymphocytes using three genes. This MANAscore program, developed by the authors of the **GSE176021** dataset, has been recently accepted by *Nature Communications* (in press) [4] and further underscores the functional relevance of these genes in tumor-reactive T cells.

References for this comment:

1. Caushi, Justina X et al. "Transcriptional programs of neoantigen-specific TIL in anti-PD-1-treated lung cancers." *Nature* vol. 596,7870 (2021): 126-132. doi:10.1038/s41586-021-03752-4
2. van der Leun AM, Thommen DS, Schumacher TN. CD8+ T cell states in human cancer: insights from single-cell analysis. *Nat Rev Cancer*. 2020;20(4):218-232. doi:10.1038/s41568-019-0235-4
3. Bassez, Ayse et al. "A single-cell map of intratumoral changes during anti-PD1 treatment of patients with breast cancer." *Nature medicine* vol. 27,5 (2021): 820-832. doi:10.1038/s41591-021-01323-8
4. Zeng Z, et al. "A minimal gene set characterizes tumor-reactive TIL specific for multiple classes of tumor antigens across different cancer types." *Journal for ImmunoTherapy of Cancer* 2024;**12**:doi: 10.1136/jitc-2024-SITC2024.0935

Minor:

"Gene programs are typically estimated using factor analysis or NMF methods. However, these methods often face issues with identifiability and impose orthogonality constraints. NMF doesn't.". This is not true for NMF.

Response:

We thank the reviewer for pointing this out. Without any assumptions, NMF is not inherently identifiable [1-5]. Identifiability refers to the property of a model that ensures unique parameter estimates can be derived from the observed data. In other words, if a model is not identifiable, it may produce varying solutions across runs, which can make the inferred latent factors less interpretable [6,7].

Foundational work on the application of NMF to gene expression data has highlighted challenges in achieving unique solutions. For example, Brunet et al. [6] observed that the NMF algorithm may produce different solutions across runs. Recent studies have addressed this issue by employing meta-analysis approaches, which average solutions over multiple replicates (often several hundred runs) to enhance robustness and reliability [7,8].

Moreover, high-dimensional gene expression data often exhibit redundancy (collinearity), with many genes showing correlated patterns. To improve interpretability, *orthogonal regularization* is sometimes applied in NMF-based methods to enforce sparse and non-overlapping solutions [9,10]. In contrast, our method, LaGrACE, requires non-orthogonal gene programs to construct the reference network.

1. Fu, Xiao, Kejun Huang, and Nicholas D. Sidiropoulos. "On identifiability of nonnegative matrix factorization." *IEEE Signal Processing Letters* 25.3 (2018)
2. Pan, Weiwei, and Finale Doshi-Velez. "A characterization of the non-uniqueness of nonnegative matrix factorizations." *arXiv preprint arXiv:1604.00653* (2016)
3. Donoho, David, and Victoria Stodden. "When does non-negative matrix factorization give a correct decomposition into parts?." *Advances in neural information processing systems* 16 (2003)
4. Laurberg, Hans, et al. "Theorems on positive data: On the uniqueness of NMF." *Computational intelligence and neuroscience* 2008.1 (2008)
5. Huang, Kejun, Nicholas D. Sidiropoulos, and Ananthram Swami. "Non-negative matrix factorization revisited: Uniqueness and algorithm for symmetric decomposition." *IEEE Transactions on Signal Processing* 62.1 (2013)
6. Brunet, Jean-Philippe, et al. "Metagenes and molecular pattern discovery using matrix factorization." *Proceedings of the national academy of sciences* 101.12 (2004)
7. Kotliar, Dylan, et al. "Identifying gene expression programs of cell-type identity and cellular activity with single-cell RNA-Seq." *Elife* 8 (2019)
8. Pelka, Karin, et al. "Spatially organized multicellular immune hubs in human colorectal cancer." *Cell* 184.18 (2021):

9. Stražar, Martin et al. "Orthogonal matrix factorization enables integrative analysis of multiple RNA binding proteins." *Bioinformatics* (Oxford, England) vol. 32,10 (2016): 1527-35. doi:10.1093/bioinformatics/btw003
10. Esposito, Flavia et al. "Orthogonal joint sparse NMF for microarray data analysis." *Journal of mathematical biology* vol. 79,1 (2019): 223-247. doi:10.1007/s00285-019-01355-2

Better to order all main/suppl figures tables in the order they are referred to in the main text.

Response:

We thank the reviewer for pointing this out. We have reordered all main and supplementary figures and tables to align with the order in which they are referred to in the main text.

Please clarify the definition and calculation of purity scores used for benchmarking.

Response:

We adapted the concept of "purity score" from the cell type purity metric described in [1]. Purity is defined as the percentage of cells within a neighborhood that share the same label (e.g., a specific cell type).

1. **Constructing the Neighborhoods:**
We begin by creating a k-nearest neighbor (KNN) graph based on high-dimensional features (such as gene expression, gene program scores, or LaGrACE divergence scores). Each cell is represented as a node, and edges connect each cell to its k-nearest neighbors, determined using principal component analysis (PCA) on the chosen input features.
2. **Defining Neighborhood Purity:**
Each neighborhood is defined as the set of cells directly connected to a given "index cell" in the KNN graph. To calculate the purity for a given neighborhood, we:
 - Identify the most abundant label within that neighborhood.
 - Compute the purity score as the fraction of cells in the neighborhood that share this majority label.
3. **Applying to Our Dataset:**
Instead of cell type labels, we use treatment-related labels (e.g., treatment duration or drug concentration) as our "cell type" labels. By doing so, we assess whether cells exposed to the same treatment conditions cluster together and exhibit similar gene expression or gene program patterns. This allows us to measure how well a given representation (e.g., raw gene expression, gene programs, or LaGrACE features) preserves treatment-related differences.

We use the **Milo package** [1] to implement this approach. Milo provides functions to construct KNN graphs, build neighbourhoods, and compute purity scores on single-cell datasets.

We have now added this detailed explanation in the **Supplementary Materials**.

References for this comment:

1. Dann, E., Henderson, N.C., Teichmann, S.A., Morgan, M.D., & Marioni, J.C. (2022). Differential abundance testing on single-cell data using k-nearest neighbor graphs. *Nature Biotechnology*, 40, 245–253.

Ensure that the mathematical notations clearly imply a single divergence score for each gene expression program, rather than for the entire sample.

Response:

We thank the reviewer for pointing this out. We have now revised the mathematical notations on page 3 (Results section: "LaGrACE Infers Divergence Scores for Gene Program Dysregulation") and pages 11–12 (Methods section: "Pseudo Control Outcome Estimation & Divergence Estimation").

Reviewer #2:

This is a very-well written paper with detailed analysis. I have a few suggestions:

Response: We thank the reviewer for the encouragement.

1. I would suggest adding a section on computational performance, including run-time analysis and scalability for large datasets compared to small datasets. This could help readers assess the feasibility of using LaGrACE for their own datasets.

Response:

This is a great suggestion. We have added a new section on computational performance (“Scalability and runtime”) to help readers assess the feasibility of using LaGrACE for their datasets. Results are included in **Appendix Table S5**.

Briefly, the computation of LaGrACE divergence scores consists of four main steps, each with distinct computational requirements:

- Gene** **Program** **Discovery:**
We use the **LOVE algorithm** to identify latent factors corresponding to gene programs. The time complexity of this step is $O(p^2)$, where p is the number of observed features.
- Reference** **Network** **Learning:**
For learning the reference network, we employ the **FGES algorithm**. Its time complexity is also $O(p^2)$, where pp represents the number of input variables.
- Pseudo** **Control** **Outcome** **Estimation:**
This step involves fitting a **linear regression model**, with a time complexity of $O(n \cdot p^2 + p^3)$, where n is the number of observations and p is the number of variables (a variable of interest and its Markov blanket).
- Divergence** **Estimation:**
For estimating divergence, we compute **element-wise differences**, which have a time complexity of $O(n)$, where n is the number of observations.

To provide a concrete understanding of LaGrACE’s performance, we tested it on two datasets using a 3 GHz 6-Core Intel Core i5 iMac:

- scRNA-seq data of A549 cells treated with SAHA: 5,530 cells (1,005 reference samples) with 3,000 highly variable genes.
- Simulated high-dimensional data: 4,000 samples (1,000 reference samples) with 2,500 continuous and 25 discrete variables.

The run-times for each step are summarized below:

Appendix Table S5. Runtime of LaGrACE. Two datasets were tested using a 3 GHz 6-Core Intel Core i5 iMac: scRNA-seq data from 5,530 A549 cells treated with SAHA (including 1,005 reference samples) with 3,000 highly variable genes, and simulated high-dimensional data with 4,000 samples (1,000 reference samples) containing 2,500 continuous and 25 discrete variables.

Step	scRNA-seq Data (seconds)	Simulated Data (seconds)
Step 1	60.8	69.2
Step 2	5.5	5.4
Step 3 & 4	1.5	1.8

2. The paper shows the technical success of LaGrACE in identifying clusters and stratifying patients. It would be great to expand the discussion on possible applications on precision medicine and how the findings would impact treatment outcomes.

Response:

We thank the reviewer for this suggestion. We have expanded the discussion section to include potential applications of LaGrACE in precision medicine.

Reviewer #3:

In this manuscript, the authors propose a novel approach LaGrACE, which estimates regulatory network-based pseudo control outcome to characterize gene program dysregulation. Given dysregulation of regulatory network can lead to a broad range of diseases, LaGrACE facilitates grouping of samples exhibiting similar patterns of gene program dysregulation. However, before this work is deemed suitable for publication, several points require further clarification:

1. How does the author ensure that the best latent factors are learned? Are these latent factors interpretable? I mean do they correspond to some biological entities? Additionally, how is the value of k determined when calculating the factor matrix through matrix factorization?

Response:

We thank the reviewer for this comment, which gives us a chance to further clarify our algorithm. LaGrACE determines the optimal number of latent factors through a model selection process that involves tuning a key hyperparameter (denoted as δ) which controls the dimensionality of the latent factor space. To select δ , we rely on the Akaike Information Criterion (AIC). In our empirical assessments, AIC has consistently provided more reliable guidance for subtyping task than the Bayesian Information Criterion (BIC) or traditional cross-validation approaches. This procedure effectively determines the value of k , which corresponds to the number of latent factors retained in the final factorization.

In terms of interpretability, the latent factors are designed to represent underlying gene programs and they are learned from the data using the algorithm **LOVE** [1]. The value of each latent factor is calculated from the values of the observed variables that comprise this factor (the “gene program”). As such, latent factors are interpretable since associations among them or between them and clinical variables correspond to associations between these “gene programs” they represent and the clinical variables.

By performing gene set enrichment analyses on the genes most strongly linked to a given factor, we can connect that factor to known biological pathways, processes, or ontologies. This step helps interpret latent factors in a biological context, allowing them to correspond to functional entities such as signaling pathways, cell cycle regulators, or immune response modules.

References for this comment:

1. Bing, Xin et al. “Latent Model-Based Clustering for Biological Discovery.” *iScience* vol. 14 (2019): 125-135. doi:10.1016/j.isci.2019.03.018

2. In complex biological systems, there are numerous feedback mechanisms among biological molecules and processes (biological processes is one type of latent factors in this manuscript). Given this, is it reasonable to model these relationships using Directed Acyclic Graphs (DAGs)? Alternatively, in the presence of loops, is the construction of the Markov skeleton based on conditional independence still correct? It is suggested that the author conducts some derivations and proofs.

Response:

This is a great question. We fully acknowledge that complex biological systems often exhibit intricate feedback mechanisms and loops among biomolecules and processes. Such feedback loops challenge the assumption of acyclicity inherent in Directed Acyclic Graphs (DAGs).

In our work, due to the high dimensionality of omics data, we employ the Fast Greedy Equivalence Search (FGES) algorithm to infer a partially Directed Acyclic Graph (pDAG), which represents a Markov equivalence class of the underlying data-generating DAG. For the datasets employed in this study, we rely on snapshots of transcriptomic profiles obtained from control (untreated) samples. We assume that these control conditions reflect relatively stable states, where long-term or dynamic feedback loops are less frequent. Under these conditions, using a pDAG approximation is still beneficial for identifying key conditional dependencies, latent factor relationships, and potential regulatory influences. The construction of Markov blankets in this context remains a valid approach for identifying sets of variables associated with any variable of interest. Additionally, while our primary goal is to estimate alterations in the reference regulatory network, changes caused by long-term or dynamic feedback loops can be inferred from disease (treated) samples by comparing their dysregulation patterns with the control reference network.

We acknowledge that if strong feedback loops are present, pDAGs and Markov blankets may not fully capture the underlying regulatory structure. Feedback loops can lead to perfect cancellations or equilibria that are not easily discernible through simple conditional independence tests or correlation-based methods. In such scenarios, the observed conditional independencies may fail to distinguish between different underlying feedback structures. We have included this as a limitation in the discussion section of the manuscript.

To address these general limitations, alternative methods such as the Fast Causal Inference (FCI) algorithm can be employed to infer the regulatory network. FCI is designed to handle more complex causal structures, including those involving latent variables and potential feedback loops. Recent theoretical work [1] indicates that under certain assumptions (e.g., σ -separation Markov property and σ -faithfulness), the output of FCI remains sound and complete even in the presence of cycles. Unlike DAGs, FCI constructs Mixed Ancestral Graphs (MAGs). In a faithful MAG, the Markov blanket of a node X is the set of parents, children, children's parents (spouses) of X , as well as the district of X and of the children of X , and the parents of each node of these districts, where the district of a node Y is the set of all nodes reachable from Y using only bidirected edges. In future versions of the LaGrACE algorithm we will explore this possibility.

To address the comment of this reviewer, we added some text in the Discussion to highlight the potential limitation of the assumption of acyclicity may pose for LaGrACE.

References for this comment:

1. Mooij, Joris M et al.. "Constraint-based causal discovery using partial ancestral graphs in the presence of cycles." Conference on Uncertainty in Artificial Intelligence. Pmlr, 2020.

3. In the experiment of "Identifying Driving Networks of Breast Cancer Subtypes", is there a way to prove (both wet lab and dry lab experiments are acceptable) that this network indeed plays a decisive role in determining cancer subtypes?

Response:

We thank the reviewer for this question. Unfortunately, there is no wet lab experimentation to validate these subtypes. However, as highlighted in our response to Reviewer 1, the LaGrACE-defined clusters exhibit significant associations with clinically relevant features, such as survival, relapse, and disease progression, particularly for Luminal B and HER2+ breast cancer subtypes. For example:

- Luminal B samples: Clusters are significantly associated with disease-specific survival (p-value = 0.0384), distal relapse (p-value = 0.0142), and the Nottingham Prognostic Index (NPI) (p-value = 0.00014).
- HER2+ samples: Clusters are significantly associated with the NPI (p-value = 0.017).

Additionally, we identified cluster-specific factors enriched for distinct biological pathways that are significantly associated with survival:

- Cluster 1: Galactose metabolism, linked to survival and metastasis (HR = 0.58, p-value = 4.7E-04).
- Cluster 2: Integrin-Linked Kinase (ILK) signaling, associated with metastatic progression and therapeutic vulnerabilities (HR = 1.27, p-value = 7.77E-04).
- Cluster 3: E2F transcriptional targets (HR = 0.77, p-value = 0.022) and regulation of RhoA activity (HR = 1.68, p-value = 4.44E-05), both critical in tumor progression and metastasis.

These findings suggest that the pathways driving each cluster contribute to survival differences, underscoring their role in subtype-specific disease progression.

24th Mar 2025

Manuscript Number: MSB-2024-12650R

Title: LaGrACE: Estimating gene program dysregulation with latent regulatory networks

Dear Dr. Benos,

Thank you for the submission of your revised manuscript to Molecular Systems Biology. We have now received the enclosed reports from the referees that were asked to re-assess it. As you will see the reviewers are now globally supportive and I am pleased to inform you that we will be able to accept your manuscript pending the following final amendments:

- 1) Please include an institutional email address on the title page of the manuscript for the corresponding author Panayiotis Benos.
- 2) In the main manuscript file, please include keywords to max. 5.
- 3) Please remove the Code Availability section and add this information into the Data availability statement. The Data Availability statement should be reserved for new code and datasets generated in the course of the current study. For reference to publicly available datasets that were used, please include a reference to the paper and/or a data citation (see point 4). The Data Availability statement should be formatted according to the following example:
"The datasets and computer code produced in this study are available in the following databases:
- Chip-Seq data: Gene Expression Omnibus GSE46748 (<https://www.ncbi.nlm.nih.gov/geo/query/acc.cgi?acc=GSE46748>)
- Modeling computer scripts: GitHub (<https://github.com/SysBioChalmers/GECKO/releases/tag/v1.0>)
- [data type]: [full name of the resource] [accession number/identifier] ([doi or URL or identifiers.org/DATABASE:ACCESSION])"
- 4) Our journal encourages inclusion of *data citations in the reference list* to directly cite datasets that were re-used and obtained from public databases. Data citations in the article text are distinct from normal bibliographical citations and should directly link to the database records from which the data can be accessed. In the main text, data citations are formatted as follows: "Data ref: Smith et al, 2001" or "Data ref: NCBI Sequence Read Archive PRJNA342805, 2017". In the Reference list, data citations must be labeled with "[DATASET]". A data reference must provide the database name, accession number/identifiers and a resolvable link to the landing page from which the data can be accessed at the end of the reference. Further instructions are available at <https://www.embopress.org/page/journal/17574684/authorguide#authorshipguidelines>
- 5) Author contributions: Please remove it from the manuscript and specify author contributions in our submission system. CRediT has replaced the traditional author contributions section because it offers a systematic machine-readable author contributions format that allows for more effective research assessment. You are encouraged to use the free text boxes beneath each contributing author's name to add specific details on the author's contribution. More information is available in our guide to authors:
<https://www.embopress.org/page/journal/17574684/authorguide#referencesformat>
- 6) References: Please correct the reference citation in the reference list to be alphabetical (not numerical). Where there are more than 10 authors on a paper, only the first 10 should be listed, followed by "et al.". Please check "Author Guidelines" for more information.
<https://www.embopress.org/page/journal/17574684/authorguide#referencesformat>
- 7) In the Methods, please take care of the following:
 - The Materials and Methods section should be renamed to "Methods".
 - Cell lines: Please also be sure to include a sentence in the Methods as to whether or not the cell lines were recently authenticated and tested for mycoplasma contamination. Please also be sure to update the Author Checklist with this information and where it can be found in the manuscript.
 - Please ensure that a statement on whether or not blinding was done is included in the Methods even if no blinding was done. Please also be sure to update the Author Checklist with this information and where it can be found in the manuscript.
- 8) All Materials and Methods need to be described in the main text using our 'Structured Methods' format. According to this format, the Methods section includes a Reagents and Tools Table (listing key reagents, experimental models, software and relevant equipment and including their sources and relevant identifiers) followed by a Methods and Protocols section describing the methods, ideally using a step-by-step protocol format. The aim is to facilitate adoption of the methodologies across labs. Please download and fill our Reagents and Tools Table template (.docx), which you can find in our author guidelines:
<https://www.embopress.org/page/journal/14693178/authorguide#structuredmethods>.
When submitting your revised manuscript, please do not include the Reagents and Tools Table in the Methods section of the manuscript but upload it as a separate file choosing the file type "Reagent Table".
An example of a Method paper with Structured Methods can be found here:
<https://www.embopress.org/doi/10.15252/msb.20178071>.
- 9) Please place individual sections of the manuscript in the following order: Title page - Abstract & Keywords - Introduction - Results - Discussion - Methods - Data Availability - Acknowledgements - Disclosure and Competing Interests Statement - References - Figure Legends - Expanded View Figure Legends.
- 10) For the figures and figure legends, please take care of the following:
 - Please note that the exact p values are not provided in the legends of figures 3C, 5H.
 - Please indicate the statistical test used for data analysis in the legends of figures 4B, C, D, F, G, H, I; 5C, E.

- Please note that the box plots need to be defined in terms of minima, maxima, centre, bounds of box and whiskers, and percentile in the legends of figures 2B, C, E; 3C, 5B, F.
 - Please note that information related to n is missing in the legends of figures 2B, C, E; 3C, F; 5B, F, H.
 - Please note that the error bars are not defined in the legend of figure 5H.
- 11) Appendix file: In the Appendix file, please ensure the title page contains "Appendix for [manuscript title]" and the Table of Contents has page numbers for the listed items.
 - 12) Funding: Please ensure that all funding sources are entered in the manuscript submission system as well as listed in the manuscript. Currently missing in the manuscript: R01HL157879; currently missing in our submission system: the National Institutes of Health (NIH): R01HL178032, R01HL127349, R01AA028436 (PVB); P30CA047904
 - 13) Synopsis image: Please ensure that the synopsis image fits into our size parameters 550 pixels wide x (300-600) pixels high (although we can resize the image, the height is too small when set to 550 pixels wide).
 - 14) Source Data: Please ensure that a completed Source Data checklist that was previously sent to you by my colleague Hannah Sonntag is uploaded as a Related Manuscript File. This will also help us to keep track of which figures were renumbered and which are new. The Source Data files also need to be reorganized as a single source data file (zipped) per figure for main figures (all EV and/or Appendix figure Source Data can be included in a single folder), with the panels clearly visible in the folder structure instead of a single excel file for all Source Data. e.g. all the Source data files for figure 1 need to be saved in a single folder and this needs to be zipped and then uploaded as "SD figure 1.zip" file.
 - 15) As part of the EMBO Publications transparent editorial process initiative (see our policy here: https://www.embopress.org/transparent-process#Review_Process), Molecular Systems Biology will publish online a Peer Review File (PRF) to accompany accepted manuscripts. This file will be published in conjunction with your paper and will include the anonymous referee reports, your point-by-point response and all pertinent correspondence relating to the manuscript. Let us know whether you agree with the publication of the PRF and as here, if you want to remove or not any figures from it prior to publication. Please note that the Authors checklist will be published at the end of the PRF.
 - 16) After your paper is published, we will promote it on social media. If you have any handles or hashtags for Bluesky you would like included, please let us know.
 - 17) Please provide a point-by-point letter INCLUDING my comments and your detailed responses (as Word file).

I look forward to reading a new revised version of your manuscript as soon as possible.

Yours sincerely,

Poonam Bheda, PhD
Scientific Editor
Molecular Systems Biology

Reviewer #1:

Authors have adressed all my concerns and revised manuscript is suitable for publication.

Reviewer #2:

The authors responded all my (and other reviewers') comments with great detail and improved the manuscript.

Reviewer #3:

The authors have addressed all my concerns, and I recommend acceptance for publication.

POINT-BY-POINT RESPONSE TO EDITORIAL COMMENTS

Manuscript Number: MSB-2024-12650

Title: LaGrACE: Estimating gene program dysregulation with latent regulatory networks

Dear Dr. Bheda,

Thank you very much for acting as Editor for our manuscript. We were happy to see that we addressed all reviewers' comments to their satisfaction. Below we have a point-by-point response to the remaining editorial comments.

1) Please include an institutional email address on the title page of the manuscript for the corresponding author Panayiotis Benos.

Done!

2) In the main manuscript file, please include keywords to max. 5.

Done! (added after the Abstract)

3) Please remove the Code Availability section and add this information into the Data availability statement. The Data Availability statement should be reserved for new code and datasets generated in the course of the current study. For reference to publicly available datasets that were used, please include a reference to the paper and/or a data citation (see point 4). The Data Availability statement should be formatted according to the following example:

"The datasets and computer code produced in this study are available in the following databases:

- Chip-Seq data: Gene Expression Omnibus GSE46748

(<https://www.ncbi.nlm.nih.gov/geo/query/acc.cgi?acc=GSE46748>)

- Modeling computer scripts: GitHub

(<https://github.com/SysBioChalmers/GECKO/releases/tag/v1.0>)

- [data type]: [full name of the resource] [accession number/identifier] ([doi or URL or identifiers.org/DATABASE:ACCESSION])"

Done!

4) Our journal encourages inclusion of *data citations in the reference list* to directly cite datasets that were re-used and obtained from public databases. Data citations in the article text are distinct from normal bibliographical citations and should directly link to the database records from which the data can be accessed. In the main text, data citations are formatted as follows: "Data ref: Smith et al, 2001" or "Data ref: NCBI Sequence Read Archive PRJNA342805, 2017". In the Reference list, data citations must be labeled with "[DATASET]". A data reference must provide the database name, accession number/identifiers and a resolvable link to the landing page from which the data can be accessed at the end of the reference. Further instructions are available at <https://www.embopress.org/page/journal/17574684/authorguide#referencesformat>.

Not sure what this means, but we are currently listing all the GEO datasets we used in the “Data availability” section.

5) Author contributions: Please remove it from the manuscript and specify author contributions in our submission system. CRediT has replaced the traditional author contributions section because it offers a systematic machine-readable author contributions format that allows for more effective research assessment. You are encouraged to use the free text boxes beneath each contributing author's name to add specific details on the author's contribution. More information is available in our guide to authors:

<https://www.embopress.org/page/journal/17574684/authorguide#authorshipguidelines>

Done!

6) References: Please correct the reference citation in the reference list to be alphabetical (not numerical). Where there are more than 10 authors on a paper, only the first 10 should be listed, followed by "et al.". Please check "Author Guidelines" for more information.

<https://www.embopress.org/page/journal/17574684/authorguide#referencesformat>

Done!

7) In the Methods, please take care of the following:

- The Materials and Methods section should be renamed to "Methods".

Done!

- Cell lines: Please also be sure to include a sentence in the Methods as to whether or not the cell lines were recently authenticated and tested for mycoplasma contamination. Please also be sure to update the Author Checklist with this information and where it can be found in the manuscript.

We have now added the appropriate sentences.

- Please ensure that a statement on whether or not blinding was done is included in the Methods even if no blinding was done. Please also be sure to update the Author Checklist with this information and where it can be found in the manuscript.

Done!

8) All Materials and Methods need to be described in the main text using our 'Structured Methods' format. According to this format, the Methods section includes a Reagents and Tools Table (listing key reagents, experimental models, software and relevant equipment and including their sources and relevant identifiers) followed by a Methods and Protocols section describing the methods, ideally using a step-by-step protocol format. The aim is to facilitate adoption of the methodologies across labs.

Please download and fill our Reagents and Tools Table template (.docx), which you can find in our author guidelines:

The Reagents Table was completed and added to the Methods section.

An example of a Method paper with Structured Methods can be found here:
<https://www.embopress.org/doi/10.15252/msb.20178071>. "

9) Please place individual sections of the manuscript in the following order: Title page - Abstract & Keywords - Introduction - Results - Discussion - Methods - Data Availability - Acknowledgements - Disclosure and Competing Interests Statement - References - Figure Legends - Expanded View Figure Legends.

The sections are placed in the correct order. However, we only have one section with Figure legends. We have also placed the Tables right before the Figure legends.

10) For the figures and figure legends, please take care of the following:

- Please note that the exact p values are not provided in the legends of figures 3C, 5H.
- Please indicate the statistical test used for data analysis in the legends of figures 4B, C, D, F, G, H, I; 5C, E.
- Please note that the box plots need to be defined in terms of minima, maxima, centre, bounds of box and whiskers, and percentile in the legends of figures 2B, C, E; 3C, 5B, F.
- Please note that information related to n is missing in the legends of figures 2B, C, E; 3C, F; 5B, F, H.
- Please note that the error bars are not defined in the legend of figure 5H.

All these have been taken care of.

11) Appendix file: In the Appendix file, please ensure the title page contains "Appendix for [manuscript title]" and the Table of Contents has page numbers for the listed items.

Done.

12) Funding: Please ensure that all funding sources are entered in the manuscript submission system as well as listed in the manuscript. Currently missing in the manuscript: R01HL157879; currently missing in our submission system: the National Institutes of Health (NIH): R01HL178032, R01HL127349, R01AA028436 (PVB); P30CA047904

Thanks for catching this. We are now consistently reporting the funding sources.

13) Synopsis image: Please ensure that the synopsis image fits into our size parameters 550 pixels wide x (300-600) pixels high (although we can resize the image, the height is too small when set to 550 pixels wide).

We have now created an expanded version of the previous synopsis image with dimensions 2290x2370 pixels, which can be downsized as needed.

14) Source Data: Please ensure that a completed Source Data checklist that was previously sent to you by my colleague Hannah Sonntag is uploaded as a Related Manuscript File. This will also help us to keep track of which figures were renumbered and which are new. The Source Data files also need to be reorganized as a single source data file (zipped) per figure for

main figures (all EV and/or Appendix figure Source Data can be included in a single folder), with the panels clearly visible in the folder structure instead of a single excel file for all Source Data. e.g. all the Source data files for figure 1 need to be saved in a single folder and this needs to be zipped and then uploaded as "SD figure 1.zip" file.

The Author Checklist has been updated. The Source Data Checklist has been submitted as "Related Manuscript File". Independent zip files with Source Data for Figures 3 and 5 have been uploaded.

15) As part of the EMBO Publications transparent editorial process initiative (see our policy here: https://www.embopress.org/transparent-process#Review_Process), Molecular Systems Biology will publish online a Peer Review File (PRF) to accompany accepted manuscripts. This file will be published in conjunction with your paper and will include the anonymous referee reports, your point-by-point response and all pertinent correspondence relating to the manuscript. Let us know whether you agree with the publication of the PRF and as here, if you want to remove or not any figures from it prior to publication. Please note that the Authors checklist will be published at the end of the PRF.

We have no objection in publishing the PRF as is.

16) After your paper is published, we will promote it on social media. If you have any handles or hashtags for Bluesky you would like included, please let us know.

Bluesky: [@benoslab.bsky.social](https://bsky.app/profile/benoslab.bsky.social)

LinkedIn: [takisbenos](https://www.linkedin.com/in/takisbenos)

26th Apr 2025

Manuscript number: MSB-2024-12650RR

Title: LaGrACE: Estimating gene program dysregulation with latent regulatory networks

Dear Dr. Benos,

Thank you again for sending us your revised manuscript. We are now satisfied with the modifications made and I am pleased to inform you that your paper has been accepted for publication.

Yours sincerely,

Sincerely,

Poonam Bheda, PhD
Scientific Editor
Molecular Systems Biology
